



# Tracking water masses using passive-tracer transport in NEMO v3.4 with NEMOTAM: application to North Atlantic Deep Water and North Atlantic Subtropical Mode Water

Dafydd Stephenson[1], Simon Müller[1,2], and Florian Sévellec[1,3]

[1]Ocean and Earth Science, National Oceanography Centre Southampton, University of Southampton, UK
[2]National Oceanography Centre, European Way, Southampton, UK
[3]Laboratoire d'Océanographie Physique et Spatiale, Univ.-Brest CNRS IRD Ifremer, Brest, France

**Correspondence:** Dafydd Stephenson (D.Stephenson@noc.soton.ac.uk)

**Abstract.** Water mass ventilation provides an important link between the atmosphere and the global ocean circulation. In this study, we present a newly developed, probabilistic tool for offline water mass tracking. In particular, NEMOTAM, the tangent-linear and adjoint counterpart to the NEMO ocean general circulation model, is modified to allow passive-tracer transport. By terminating dynamic feedbacks in NEMOTAM, tagged water can be tracked forward and backwards in time as a passive dye,

producing a probability distribution of pathways and origins, respectively. Upon contact with the surface, the tracer is removed from the system, and a record of ventilation is produced.

Two test cases are detailed, examining the creation and fate of North Atlantic Subtropical Mode Water (NASMW) and North Atlantic Deep Water (NADW) in a 2° configuration of NEMO run with repeated annual forcing for up to 400 years. NASMW is shown to have an expected age of 4.5 years, and is predominantly eradicated by internal processes. A bed of more persistent

NASMW is detected below the mixed layer with an expected age of 8.7 years. It is shown that while model NADW has two distinct outcrops (in the Arctic and North Atlantic), its formation primarily takes place in the subpolar Labrador and Irminger seas. Its expected age is 112 years.

## 1   Introduction

The intricate process by which the atmosphere and ocean exchange properties has decisive effects on oceanic and atmospheric circulation, biochemistry and climate. Tracing the pathways of newly ventilated ocean water masses is a multidisciplinary pursuit, for which an entire toolbox of methods has been developed.

Thermocline ventilation and water mass formation is an area with a long history of inquiry. From charts of hydrographic sections, Iselin (1939) inferred communication between isopycnals at depth and their outcrops. This idea was developed theoreti-

cally over the following decades in analytical models (Welander, 1959, 1971; Luyten et al., 1983). In this vertically-discretised,





laminar framework, potential vorticity (PV) conservation facilitates a bijective relationship between the origin and destination of a water parcel. In reality, however, turbulent mixing means that surface-to-deep pathways have a pronounced probabilistic character.

Observational studies have been able to capture snapshots of this turbulent behaviour locally in passive-tracer dye-release experiments (e.g. Schuert, 1970). However, the effect of turbulence on water mass pathways has only been observed on a large scale since the widespread adoption of cost-effective Lagrangian profiling floats, which have been observed to follow chaotic trajectories (e.g. Fischer and Schott, 2002; Fratantoni et al., 2013; Bower et al., 2019). Despite the number of these floats, these pathways remain under-sampled.

Simulations of larger numbers of Lagrangian particles in sophisticated eddy-resolving ocean general circulation models (OGCMs, e.g. Blanke and Raynaud, 1997; Gary et al., 2014) are reworking long-held assumptions about the routes taken by newly formed water masses. Despite these new developments, such studies are limited by the large computational expense of eddy-resolving models, and the fact that sub-grid-scale dispersion of Lagrangian particles is typically not parameterised. An alternative approach is to use a simulated passive tracer (c.f. England and Maier-Reimer, 2001). Such tracers are spatial distributions of dye concentrations which undergo diffusion. Lagrangian particles, conversely, are indivisible nodules which may only be advected by the immediate flow. As such, an infinite number of them is required to fully represent an equivalent dye concentration.

This study presents a method for tracking water masses by means of passive-tracer deployment in the tangent-linear and adjoint model (TAM) developed for the NEMO OGCM (Madec, 2012; Vidard et al., 2015). TAMs are typically used for sensitivity studies and data assimilation (Errico, 1997). However, due to its probablistic nature, it offers an advantage over the more conventional tracking technique of Lagrangian particle modelling. Within the Lagrangian framework, ocean sensitivity to initial conditions means that a very large ensemble of initially close particle deployments may be required to representatively sample the full space of particle trajectories. A TAM can be exploited to bypass this requirement, producing a continuous probability distribution of all possible particle trajectories. As such, the method does not describe particle locations and deterministic trajectories, as a Lagrangian approach would, but tracer concentrations and probabilitic pathways. The adjoint and tangent-linear of the model can respectively track the origins and fate of passive-tracer-tagged water in this manner. This native ability to track water both forward and backward in time, along with the ability to re-use a single "trajectory" run of a nonlinear model, offer an advantage over passive tracer tracking in a nonlinear OGCM.

TAM construction is typically a laborious process (e.g. Giering and Kaminski, 1998). The Jacobian of a highly involved nonlinear model (often with millions of degrees of freedom) must be computed with respect to the ocean state. This provides a linear function mapping small perturbations to the ocean state to their future outcome (the tangent-linear). The adjoint of this linear operator provides the sensitivity of the ocean state to earlier perturbations.

Several studies have bypassed such complications by computing approximations to the true adjoint of tracer transport in an OGCM. An early application of an adjoint approximation to the tracking of water masses was developed by Fukumori et al. (2004), who tracked eastern equatorial Pacific water with the Massachusetts Institute of Technology GCM. Their approximation capitalises on the comparatively simple nature of passive-tracer transport in isolation. For passive tracers, advection and


diffusion are respectively skew-symmetric and self-adjoint. They were thus able to map possible past locations of the water mass by reversing the velocity field without changing the diffusion tensor, similarly to a true passive tracer adjoint. The method is further used by Qu et al. (2009), Gao et al. (2011) and Gao et al. (2012) to track Pacific waters and Qu et al. (2013) to track the salinity maximum of the North Atlantic subtropical gyre.

Another pseudo-adjoint passive tracer approach is presented by Khatiwala et al. (2005). Rather than deriving the Jacobian of an OGCM, they derive a tangent-linear approximation empirically. The procedure constructs a "transport matrix" row-by-row by repeatedly perturbing the nonlinear model along basis vectors of the ocean state. The responses following a single time step can be synthesised to derive the elements of the matrix. While more approximate than the Fukumori et al. (2004) approach, it is also more computationally efficient by design. This allows the transport matrix method to be used for long-term passive
tracer experiments, for example simulations of long half-life radioisotopes.

The work presented herein is of a similar nature to the above studies, but repurposes an existing TAM for passive tracer tracking. In this sense its adjoint is the true adjoint of the nonlinear model, rather than a bespoke approximation.

We demonstrate the efficacy of the development through case studies of two climatically important water masses of the North Atlantic, whose formation regions are closely aligned with major components of the Atlantic Meridional Overturning
Circulation (AMOC). These are North Atlantic Subtropical Mode Water (NASMW) or Eighteen Degree Water, formed in the vicinity of the Gulf Stream, and North Atlantic Deep Water (NADW), formed in the subpolar North Atlantic.

NASMW is the name given to a homogeneous body of water found in the upper region of the western subtropical gyre (STG). It was first identified as early as the Challenger expedition, where repeat soundings revealed its unusual uniformity (Thomson, 1877, their plate XI). It was later named Eighteen Degree Water for its characteristic temperature by Worthington
(1959), who provided a first formal definition of the water mass.

NASMW formation has historically been attributed to surface heat loss during winter (e.g. Worthington, 1972; Speer and Tziperman, 1992) although mechanical forcing has been proposed to have an effect on formation by destroying PV (e.g. Thomas, 2005). Formation rate estimates by varying methods have produced substantially different results (addressed by Marshall et al., 2009), and it has recently been suggested that the nature of formation is location-dependent (Joyce, 2011).

Newly ventilated NASMW follows the subtropical gyre circulation, travelling north with the Gulf Stream (Klein and Hogg, 1996). Data from profiling floats (Kwon and Riser, 2005; Fratantoni et al., 2013) and Lagrangian particle modelling (Gary et al., 2014) suggest that a minority of this water should then be exported to the subpolar gyre.

Multiple studies have sought to quantify or describe the fate of NASMW. Forget et al. (2011) estimate that atmospheric exchange is roughly twice as effective at destroying existing NASMW as ocean internal mixing. Davis et al. (2013) discern
between NASMW removal by air–sea exchange and internal mixing as respectively fast and slow processes, which occur at different stages of its seasonal cycle. They argue for the existence of a persistent reservoir of NASMW below the mixed layer, which is shielded from destruction by a layer of high PV. Gary et al. (2014) use a Lagrangian modelling approach to analyse the final stages of the NASMW life cycle, and show that the nature of NASMW destruction depends on how the water mass is defined - the lack of a universally accepted definition of NASMW commonly leads to such conflicts between studies (Joyce,
35   2011).





A quite different water mass in nature, NADW is one of the two primary high-density water masses formed in polar waters which act as pre-conditioners of the thermohaline component of the AMOC (Sévellec and Fedorov, 2011). NADW formation involves a range of processes and source water types. The dominant contributors are Labrador Sea Water (LSW, e.g. Talley and McCartney, 1982) and denser Overflow Waters (OW, e.g. Swift et al., 1980; Hansen et al., 2001), which respectively contribute

to lower and upper NADW. LSW is created by cooling and convection of mode waters (McCartney and Talley, 1982) in the Labrador and Irminger Seas (Pickart et al., 2003a, b; Jong and Steur, 2016) during sufficiently severe winters (Clarke and Gascard, 1983). Meanwhile, OW forms as warm North Atlantic water crosses the Greenland-Scotland ridge, cools and sinks (Quadfasel and Käse, 2007). Resulting pressure gradients drive OW back over the ridge into the Atlantic at depth (Hansen and Østerhus, 2000).

Both source waters are exported southward following formation. The classical view from in-situ current measurements (e.g. Leaman and Harris, 1990; Molinari et al., 1992) is that the main southward pathway is the deep western boundary current (DWBC). However, new data from profiling floats and modelled Lagrangian drifters (Bower et al., 2009, 2011) suggest that this is not the complete story. Gary et al. (2011) investigate alternative export pathways in a hierarchy of model resolutions. They find a second, slower pathway driven by a series of deep, eddy-driven anticyclonic gyres. These facilitate the southward

transport of water parcels which have become detached from the DWBC, but are only found in models which are of eddy-resolving resolution. At lower-resolution, Lagrangian particles tend to follow the DWBC (Straneo et al., 2003).

Beyond the Equator, NADW mixes laterally and vertically with ambient water masses, eventually entering the Indian and Pacific basins via the Antarctic Circumpolar Current (Reid and Lynn, 1971). In these basins, it retains a thickness of several hundred metres (Johnson, 2008), and reaches an estimated age of several centuries (e.g. Broecker, 1979; Hirst, 1999).

Here, we present a new tool which allows us to track water masses to and from their point of formation, and apply it to NASMW and NADW. We demonstrate the use of this development by considering the life cycles of the above water masses in a new framework. The paper is set out as follows. In Sect. 2, we summarise the mathematical theory of the TAM approach. We then describe the model, our modifications to it, and how these may be applied to water mass tracking. The application of this development to NASMW is discussed in Sect. 3 with similar considerations for NADW in Sect. 4. We conclude in Sect. 5

with a summary of our method and findings, highlighting any recommendations for future work.

## 2   Development of the passive tracer module

### 2.1   Mathematical background

The tangent-linear method describes the evolution of perturbations to the ocean state in a linear framework. Perturbations are described by vectors following the structure of the ocean state vector, containing values for all prognostic variables at all

locations. The linear evolution of an initial perturbation $|\boldsymbol{u}_0\rangle$ to the ocean state vector is described by the equation

$$|\boldsymbol{u}(t)\rangle = \boldsymbol{\Psi}(t, t_0)|\boldsymbol{u_0}\rangle \qquad\qquad (1)$$





where $|\boldsymbol{u}(t)\rangle$ provides the condition of the perturbation at time $t$, $\boldsymbol{\Psi}(t, t_0)$ is known as the propagator matrix and we have used the "bra-ket" notation of Dirac (1939).

The adjoint approach considers the sensitivity of properties of interest to earlier perturbations. For a "cost function" $J$ (mapping the state vector to such a property) which is scalar-valued and linear, that is $J(\boldsymbol{u}) = \langle \mathbf{F} | \boldsymbol{u} \rangle$ (for some $\mathbf{F}$), the sensitivity is given by

$$\frac{\partial J}{\partial |\boldsymbol{u_0}\rangle} = \boldsymbol{\Psi}^{\dagger}(t_0, t) |\mathbf{F}\rangle \tag{2}$$

where $\boldsymbol{\Psi}^{\dagger}(t_0, t)$ is the adjoint of the propagator matrix, correspondent to its transpose in a Euclidean inner product space. These relationships are derived in full by Errico (1997). Furthermore, we have

$$\langle \mathbf{F} | \boldsymbol{\Psi}(t, t_0) | \boldsymbol{u_0}\rangle = \langle \boldsymbol{u_0} | \boldsymbol{\Psi}^{\dagger}(t_0, t) | \mathbf{F}\rangle \tag{3}$$

following the definition of the adjoint as a linear operator.

## 2.2 Passive tracer implementation and model description

Perturbations $|\boldsymbol{u}_0\rangle$ such as in Eq. 1 typically induce an active, or dynamic response (Marotzke et al., 1999). For temperature or salinity perturbations, this corresponds to a modified density field evoking changes in streamlines by modifying pressure gradients. This occurs in tandem with the passive advection and diffusion of the perturbed tracer. We may represent this by splitting the propagator into active (A) and purely passive (P) components $\boldsymbol{\Psi}(t, t_0) = \boldsymbol{\Psi}_A(t, t_0) + \boldsymbol{\Psi}_P(t, t_0)$. Note that we use the term "purely passive" for $\boldsymbol{\Psi}_P$ because the active component $\boldsymbol{\Psi}_A$ encompasses all dynamic interaction between the unperturbed ocean state and the perturbed field. This itself includes a partially passive element (for example passive transport of dynamically modified tracers). The purely passive component $\boldsymbol{\Psi}_P(t, t_0)$, on the other hand, involves no such interactions.

In the case of propagating a passive tracer, $\boldsymbol{\Psi}_A(t, t_0)$ vanishes. Our perturbation is now merely an initial injection of dye of a specified concentration, $|\boldsymbol{c}_0\rangle$, and so Eq. 1 collapses to

$$|\boldsymbol{c}(t)\rangle = \boldsymbol{\Psi}_P(t, t_0) |\boldsymbol{c}_0\rangle \tag{4}$$

This dye is unable to feed back on the ocean state, and so as a perturbation is trivial. We will hence refer to initial perturbations as "injections" and evolved perturbations as dye concentrations, as is more appropriate for the passive tracer case. The common term "cost function", which relates to optimisation problems, is also a misnomer in the more simple context of propagating passive tracers. In our analysis, $\langle \mathbf{F} |$ acts on the concentration vector $|\boldsymbol{c}(t)\rangle$ to produce a tracer budget. As such we will more suitably refer to cost functions as budget co-vectors, $\langle \mathbf{B} |$.

Hence, in summary, the tangent-linear model $\boldsymbol{\Psi}_P$ describes the time evolution of dye concentrations $|\boldsymbol{c}(t)\rangle$ in response to initial dye injections $|\boldsymbol{c}_0\rangle$. Accordingly, the adjoint model $\boldsymbol{\Psi}_P^{\dagger}$ describes the sensitivity of tracer budget co-vectors $\langle \mathbf{B} |$ to earlier dye injections $|\boldsymbol{c}_0\rangle$.

The model used herein is the NEMO 3.4 OGCM (Madec, 2012) with the tangent-linear and adjoint package NEMOTAM (Vidard et al., 2015). In particular, the former (nonlinear model) provides the unperturbed, nonlinear model trajectory. Meanwhile the latter (linear model) provides a model implementation of the propagator and its adjoint. Both models are used in the



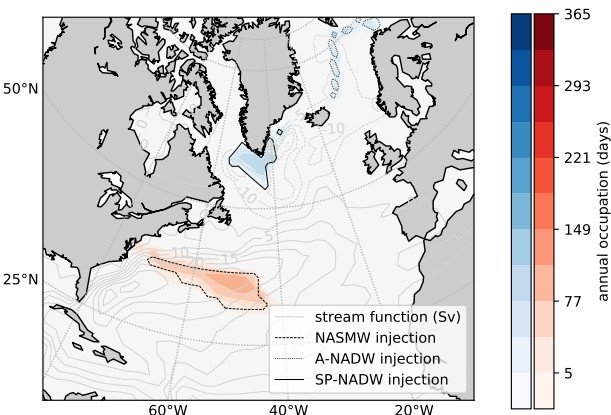

**Figure 1.** Outcrop locations of NASMW (red) and NADW (blue) from a 60 year climatology of the nonlinear model, following their respective definitions in Sections 3 and 4. Shading indicates the number of days of the year when the water mass is found at a given latitude and longitude. Light grey contours show the time-mean barotropic stream function for the North Atlantic. The dashed (solid, dotted) contour shows the distribution of the dye injection which initialises the tangent-linear run for NASMW (SP-NADW,A-NADW) at the annual maximum (see Section 3.2 (4.2.1,4.2.2))

ORCA2-LIM configuration. This consists of a 2° grid, with refinement of the equatorial and Mediterranean regions (Madec and Imbard, 1996). There are 31 vertical levels, whose height follows a hyperbolic tangent function of depth, ranging from 10 m at the surface to 1000 m. There are two ice layers (Fichefet and Maqueda, 1997) in the background state, although these are not incorporated into NEMOTAM. The model is forced using the single, repeating normal year of the CORE forcing pack-

5 age (Large and Yeager, 2004). We therefore consider our results in a climatological, as opposed to historical, context. In this configuration, model NASMW (see Section 3 for more details on the definition and properties of NASMW in the nonlinear model) most persistently outcrops in the neighbourhood of the time-mean North Atlantic barotropic stream function maximum, which is 38.8 Sv (Fig. 1, red shading and grey contours). Below the surface, it occupies a narrow latitudinal band at depths of up to 240 m (Fig. 2, red shading). Conversely, model NADW (as defined and described in Section 4) most often surfaces

in the region close to the barotropic stream function minimum of -25.3 Sv (Fig. 1, blue shading and grey contours). At most latitudes, NADW occupies depths below the local time-mean meridional stream function maximum (Fig. 2, blue shading and grey contours). The overall time-mean North Atlantic meridional stream function maximum is 16.1 Sv, occurring at 42°N at a depth of 870 m.

To isolate the purely passive response detailed in Eq. 4, we set velocity and sea surface height (SSH) modifications to zero

in the NEMOTAM time stepping procedure (Fig. 3). Further feedbacks involving parameterisations such as vertical and eddy-induced mixing are already absent in NEMOTAM due to approximations in linearisation (Vidard et al., 2015) and so did not require further modification. Additionally, eddy-induced advective velocities, although present in the nonlinear trajectory, were not included for our passive tracers in NEMOTAM.



**Figure 2.** Zonally averaged distribution of NASMW (red) and NADW (blue) in a 60 year climatology of the nonlinear model, following their respective definitions in Sections 3 and 4. Shading indicates the number of days of the year when the water mass is found at a given latitude and depth. Light grey contours show the meridional stream function of the Atlantic. Dashed contours show the distribution at the point of the year when the adjoint run is started

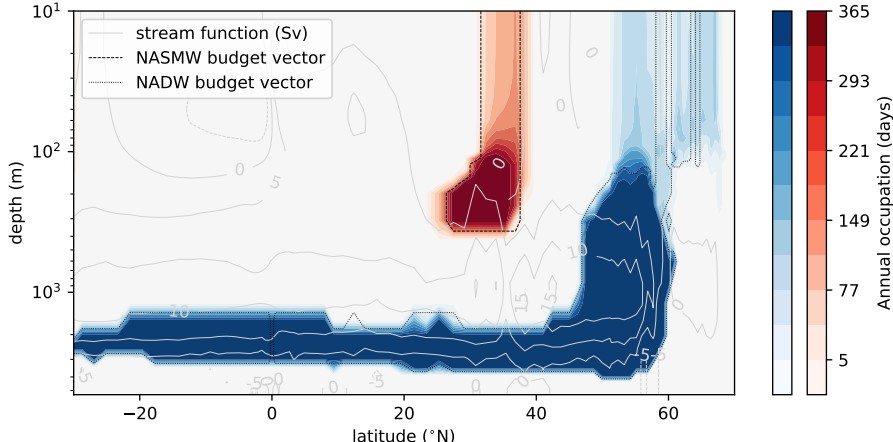

Our water mass tracking procedure is as follows. We begin by defining the water which we intend to track, for example using a temperature-salinity (TS) range. To track the evolution of newly formed water of this type, we identify all surface locations where T and S properties meet our definition in the nonlinear model trajectory. We then inject into the tangent-linear model a dye concentration of 1 in these locations, and run the model forward. To identify the origins of existing water using the adjoint

model, the procedure is similar. We begin by identifying model grid cells in the nonlinear trajectory (at all depths) which satisfy our TS definition. The budget vector is populated using the volume of these grid cells, with zero values elsewhere. This budget vector is then propagated back in time using the adjoint model to produce a sensitivity of the budget to earlier concentrations. In both cases, we remove any tracer which reaches the surface using the model's surface temperature restoring scheme. This scheme restores concentrations towards zero with a time scale of 60 days for a 50 m mixed layer (Madec, 2012). In the adjoint

case, a record of this restoring leads to a spatiotemporal probability distribution of the surface origins of the tracer.

### 2.3 Advection schemes

The default advection scheme of the nonlinear model, used to compute the background state around which the model is linearised, is a total variation diminishing (TVD) scheme. This is a flux-corrected transport scheme (FCT, Lévy et al., 2001), which balances an upstream scheme with a centred scheme using a nonlinear weighting parameter described by Zalesak (1979).

This scheme is preferable to others available in the model for non-eddying configurations (Lévy et al., 2001), but its nonlinearity means that it is not suitable for a TAM. As such, the standard advection scheme of NEMOTAM is a linearised counterpart in the form of a second-order forward-time-centred-space (FTCS) finite difference scheme. A caveat of such schemes is the presence of an negative artificial diffusion coefficient, which can lead to negative quantities of positive-definite tracers (Owen, 1984).





**Figure 3.** Adaptation of NEMOTAM time-stepping procedure for the tangent-linear (blue) and adjoint (red) cases. Steps outlined in green are new additions to the scheme.

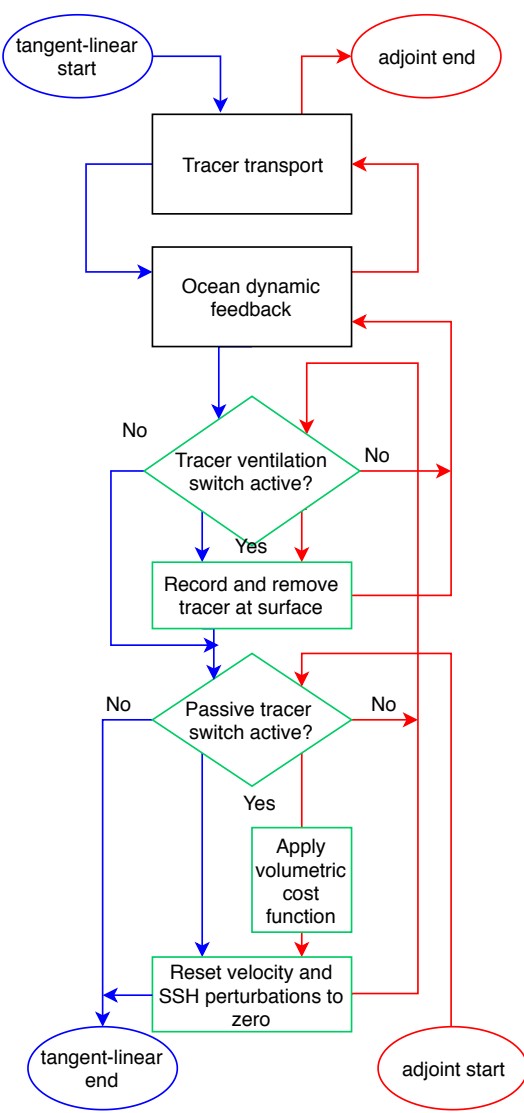

This encourages down-gradient diffusion of otherwise zero concentrations, acting as a source of further negative tracer when combined with the surface restoring scheme. This is particularly problematic in the case of passive-tracer advection as, unlike in the fully active TAM, negative quantities are unphysical.

An alternate scheme which is first-order linear (thus compatible with tangent-linear models and added to NEMOTAM for this study) is the trajectory-upstream scheme. This is an approximation to the classical upstream scheme, whereby the upstream direction is inferred from the trajectory velocity field (avoiding nonlinear sign functions). As the velocity field cannot be modified in the passive tracer case, however, the method is everywhere equivalent to the classical scheme. The scheme is





forward in both time and space, and offers the advantage of not producing negative tracer values. Nevertheless, this scheme also possesses an artificial (positive-valued) diffusion term, leading to a disproportionate spread of tracer, particularly in the vertical.

To strike a balance between the relative merits and caveats of the linear schemes described above, an approximation to the weighted-mean scheme of Fiadeiro and Veronis (1977) was coded into NEMOTAM. Like the TVD scheme, the true weighted-mean scheme is a linear combination of the classical upstream and FTCS methods. As before, we approximate the classical upstream method by using the trajectory-upstream scheme. The weighting parameter is determined dynamically from the relative strength of local transport by advection vs. diffusion in the trajectory. Despite inheriting issues from both FTCS and the trajectory-upstream schemes, it is preferable to either individually. Its negative tracer concentrations are less persistent than those of the TVD and FTCS schemes, and its anomalous vertical spread was found to be less than half that of the upstream scheme (Fig. 4). Some issues have been raised with multivariate and time-dependent extrapolations of the weighted-mean scheme (e.g. Gresho and Lee, 1981). However, nonlinear flux-corrected transport schemes are required in order to improve on the weighted-mean scheme (Gerdes et al., 1991), and these are not compatible with the linear model. We thus proceed using our approximation to the weighted-mean scheme (in both the horizontal and vertical) for the water mass case studies detailed in this paper. We remark that it is entirely possible to incorporate a nonlinear advection scheme into NEMOTAM, but that the resulting quasi-linear model would not have a true adjoint, which we require for our study.

## 3 Application to North Atlantic Subtropical Mode Water

### 3.1 NASMW definition and properties

We now apply the developments of Sect. 2 to the problem of tracking the origins and fate of NASMW in the ORCA2-LIM simulation. There is no universally accepted definition of NASMW, with differing definitions leading to conflicting results between studies (Joyce, 2011). Here, we define NASMW as water lying in the temperature band $[17, 19]°$ C, as in the original definition of Worthington (1959), with salinity in the range of $[36.4, 36.6]$ psu. As in some other studies (e.g. Kwon and Riser, 2004; Gary et al., 2014), we impose a third criterion, in our case that NASMW has a thickness of at least 125 m. This condition ensures homogeneity and isolates the water mass from the adjacent Madeira Mode Water (MMW) to its east (Siedler et al., 1987).

Model NASMW has a consistent outcrop in a single area (Fig. 1) and does not extend far south of its outcrop region below the surface (Fig. 2). The outcrop area of NASMW peaks at the beginning of April at $1.3 \times 10^6$ km$^2$ (Fig. 5), coinciding with peaks in volume ($4.2 \times 10^5$ km$^3$) and thickness (310 m) of the water mass. These maxima occur a month before the annual maximum local MLD of 430 m. Following this, rapid stratification due to summer warming shoals the mixed layer. The outcrop area concurrently diminishes until the water mass is completely sheltered from air–sea exchange. This period of submergence extends from early June to early December, as the water mass' upper surface progressively deepens to a maximum of 101 m. The volume also decreases, towards $1.3 \times 10^5$ km$^3$ at the annual minimum.



**Figure 4.** Comparison of advection schemes in NEMOTAM. To assess the schemes, one of the runs of this study (the tangent-linear run of Sect. 3.2) was repeated with four different schemes and the tracer distributions were compared. Top panel: Lateral spread (spatial standard deviation around mean tracer position) as a ratio to the corresponding spread in the nonlinear TVD scheme, Middle panel: vertical spread (as above). Bottom panel: ratio of volumes occupied by negative and positive tracer concentrations, respectively.

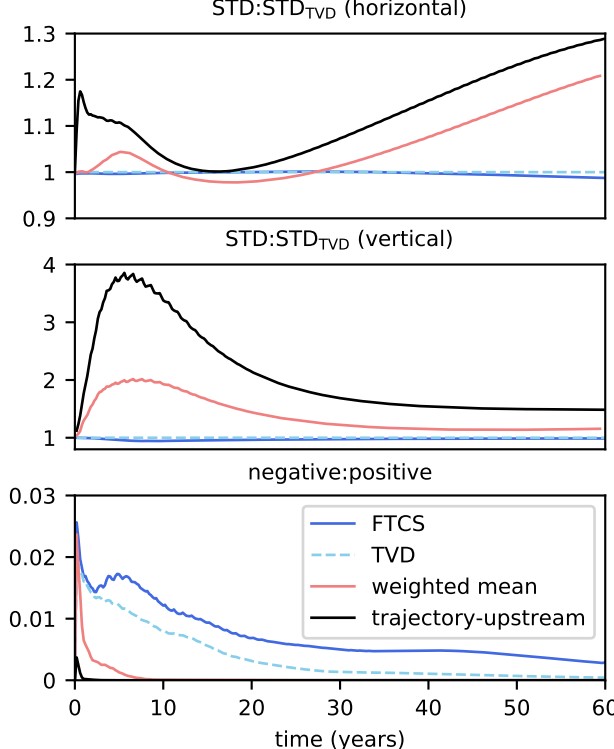

### 3.2 Tangent-linear run

As outlined in Sect. 2.2, we begin by identifying all NASMW in the surface layer of the nonlinear model at a given time. We choose this to be the time when the outcrop is maximal (Fig. 5). We then inject a concentration of 1 into the tangent-linear model at these locations, and propagate it for 60 years. Newly ventilated NASMW initially resides close to the surface (Fig. 6). Here, its behaviour is governed by surface currents, which fractionate the water mass. At the point of application, 37% of the tracer lies in the Gulf Stream. Most of this water follows the recirculation of the subtropical gyre, while a minority is exported to the subpolar region (2.5% over four years). Our quantification of subpolar export is on the same order of that found under similar conditions by Burkholder and Lozier (2011), who analyse inter-gyre exchange in a purpose-built Lagrangian study.

NASMW is short-lived. Persistent proximity to the surface leads to the re-ventilation of 95% of the initialised tracer over the 60 year run. Of that which remains in the ocean, 90% is transformed and no longer fits the definition of NASMW (Fig. 7). As time progresses, because of vertical mixing together with near-surface tracer removal by the restoring scheme, remaining tracer mechanically tends to reach deeper, colder, fresher waters. Using the passive tracer approach, it is not possible to establish a full,





**Figure 5.** Mean-year time series of NADW (blue) and NASMW (red) volume (top panel) and outcrop surface area (bottom panel) including those of the Arctic (dashed line) and subpolar (dotted line) NADW outcrops from the nonlinear trajectory. Circles indicate the injection time of NASMW (red), NADW (blue) and A-NADW (black) in the forward simulations (see Sections 3.2 and 4.2)

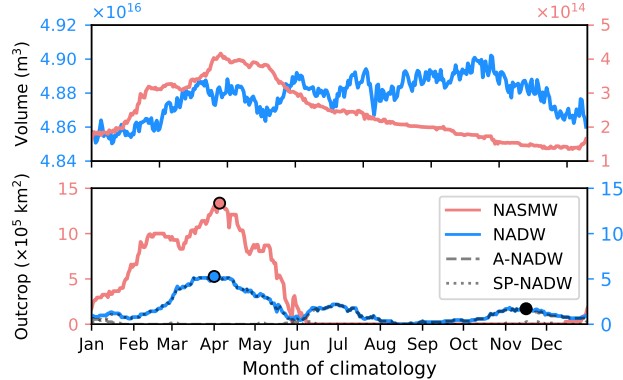

qualitative description of the fate of NASMW. However, it can be observed that only around 5% of all NASMW re-ventilation occurs within its surface outcrop. The rest is mixed out of the TS class, either remaining in the system for the remainder of the 60-year simulation, or (more commonly) resurfacing elsewhere.

Due to rapid re-exposure, the e-folding time of our NASMW is just 60 days. This is shorter than the estimations of Gary et al. (2014) (1 yr) and Kwon et al. (2015) (3 yr), but this is likely due to inherent differences in formulation. In the above studies, Lagrangian particles released in NASMW are not removed upon contact with the surface, so as to account for re-ventilation. Accounting for re-exposure, Fratantoni et al. (2013) suggest around 75 days, more closely aligned with our own findings.

## 3.3 Adjoint runs

To track existing NASMW back to its source locations, we again follow the procedure outlined in Sect. 2.2. This consists of taking a budget of NASMW in the final year of a nonlinear simulation and propagating this budget backward using the adjoint model. We begin the adjoint run at the same point in the annual cycle as the tangent-linear run, when NASMW volume is at its maximum (Fig. 5). We bisect the mode water budget co-vector $\langle \mathbf{B}_{MW}|$ into tracer above the mixed layer depth ($\langle \mathbf{B}_{MW}^{U}|$) and tracer below ($\langle \mathbf{B}_{MW}^{L}|$), and propagate each part separately. This allows us to explore the potential for a resilient layer of older NASMW below the MLD (Davis et al., 2013). By linearity, the union of the propagated budget vectors is equivalent to the propagated budget vector of NASMW as a whole: $\mathbf{\Psi}^{\dagger}(t_0, t)(|\mathbf{B}_{MW}^{U}|\rangle + |\mathbf{B}_{MW}^{L}|\rangle) = \mathbf{\Psi}^{\dagger}(t_0, t)|\mathbf{B}_{MW}\rangle$

Most NASMW propagated with the adjoint model reaches the surface quickly; 70% of the tracer-tagged water mass is under 5 years old. During this early stage, ventilation occurs predominantly within the subtropical gyre recirculation, in the neighbourhood of the NASMW outcrop (Fig. 8). The 60-year mean formation location of NASMW is also within this region, at 32° N, 58° W. This is almost coincident with the core of NASMW formation determined by Warren (1972), and agrees with the air–sea exchange-based estimate of Worthington (1959).



**Figure 6.** Evolution of NASMW-tagged tracer in the tangent-linear model. Black and white shading indicates depth-integrated probability density (corresponding to the likelihood that NASMW is found in that region) at 1, 2, 5, 10, and 50 yr. Inset percentages show the global integral of this field, i.e. the total proportion of tracer not yet re-ventilated at the surface. The coloured line tracks the centre of mass of the tracer from initialisation to its current position, with colours indicating the mean depth of the tracer. The purple contour shows the distribution of the original tracer injection.

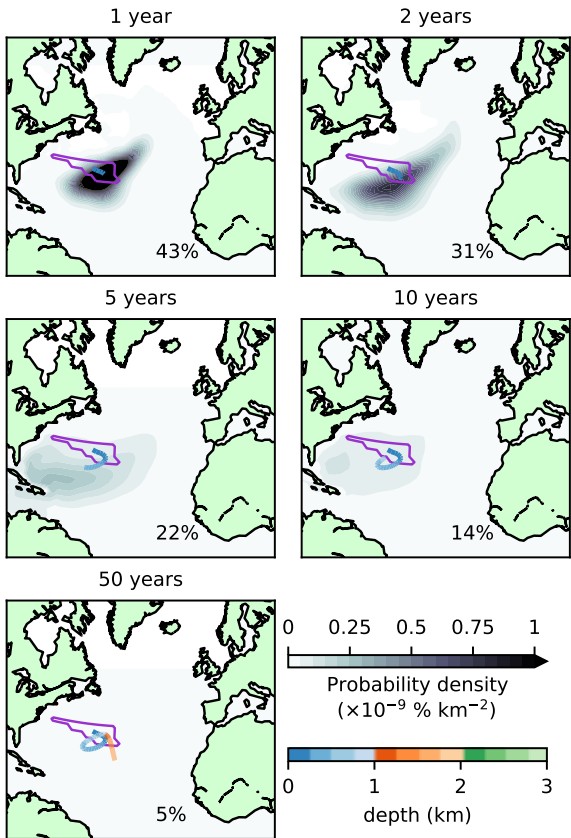

For NASMW over 5 years old, current patterns begin to have a distinct influence on formation. There is a clear signature of the Gulf Stream on the youngest NASMW. This evolves backwards as the adjoint propagates, eventually leaving an imprint of the entire subtropical gyre (Fig. 8). Simultaneously, newly formed mode water in the eastern North Atlantic (MMW) begins to cross into NASMW within 10 years. The signature of the Mediterranean outflow is particularly strong on the very oldest NASMW, which also contains contributions from the subpolar region. The culmination of all of these water types throughout the 60-year evolution is evident when the sources of NASMW are viewed in TS space (Fig. 9). Also apparent is that the primary source of NASMW has a warmer signature, which reflects the advection of warmer waters from the south, which cool and subduct to form the water mass. This suggests that the NASMW surface outcrop is not, in fact, the dominant origin of

**Figure 7.** Evolution of NASMW in TS space. Shading indicates likelihood that the water mass found in a particular TS class after 1, 2, 5, 10 and 50 yr. The red box marks the TS range used to define the water mass in this study. Contours show the density at the average depth level of the tracer.

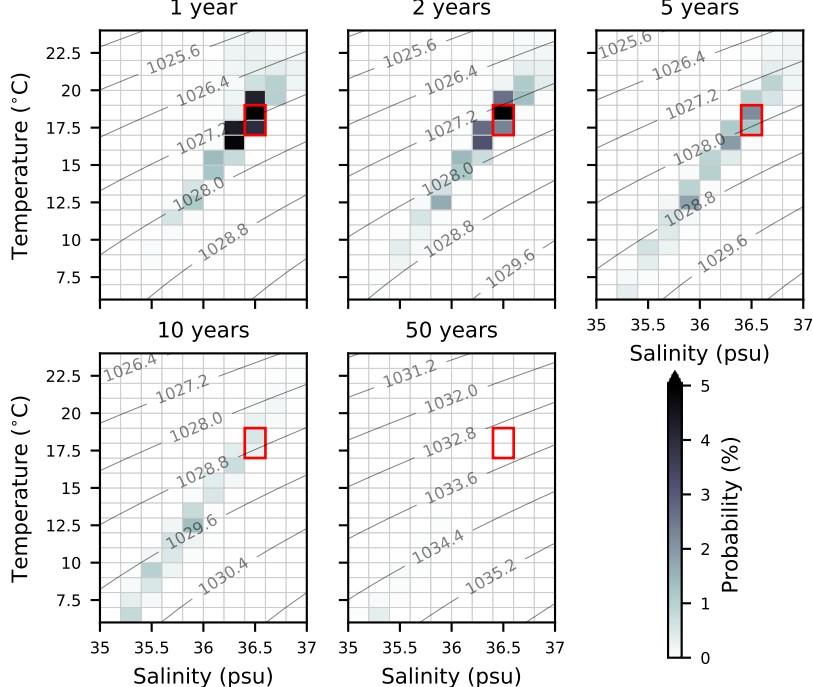

NASMW, contrary to the intuition provided by laminar models of the ventilated thermocline (e.g. Luyten et al., 1983), and that ocean dynamics play a fundamental role in mode water formation.

We also consider the time scales involved with NASMW formation. By recording the time at which tracer is removed from the budget by the surface restoring scheme, we may construct a probability distribution function (PDF) of water mass age

5  (Fig. 10). This distribution resembles a log-normal PDF with a lower skewness for NASMW lying below the MLD (Fig. 10, blue bars). Also clear is the seasonal cycle of shielding brought on by summer stratification, with less NASMW reaching the surface during these periods. Due to the simulation beginning at the annual maximum in NASMW production, peak NASMW formation occurs almost instantly. For tracer below the mixed layer (3.78% of the total) the peak does not occur until the outcrop maximum of the third year. From the PDFs, the expected age of the NASMW constituents above and below the MLD

10  are, respectively, 4.70 yr and 8.79 yr. It is evident that NASMW located below the MLD is shielded from renewal, with nearly double the expected age of NASMW as a whole.

We finally address the asymptotic tail of the PDF, which represents the oldest waters found within NASMW. Our findings suggest a high-latitude source makes up a large fraction of this water (Fig. 8). Indeed, we find that for the 30-50 yr period, some

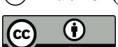

**Figure 8.** Surface origins of NASMW as determined by the backtracking budget analysis (adjoint model simulation). Shading indicates probability density. This corresponds to the likelihood that NASMW is formed in a given region during the time periods [0,5] yr, (5,10] yr, (10,30] yr and (30,50] yr. Inset percentages show the global integral (that is, the total proportion of the budget which is formed during each time period).

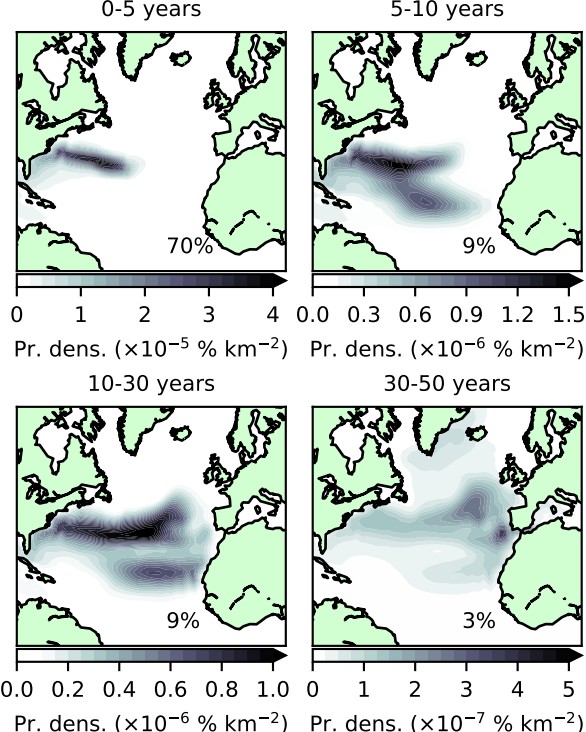

68.3 % of NASMW originates at latitudes of $45°$ or higher. The idea of a distant source of the oldest NASMW is concurrent with the few studies which have considered it. Douglass et al. (2013) use an ideal-age tracer to construct a histogram similar to our own, acknowledging a remote source of the very oldest NASMW. Kwon et al. (2015) find that at least 20% of NASMW stems from other regions, and highlight the potential importance of subpolar latitudes.

## 4 Application to North Atlantic Deep Water

### 4.1 NADW definition and properties

As in Sect. 3.1, we now consider the properties and behaviour of NADW in the ORCA2-LIM simulation. We define NADW to fall within a temperature-salinity band of $[2, 4]$ °C, $[34.9, 35.0]$ psu, in close alignment with the definition of Worthington and Wright (1970). Analysis of the nonlinear model trajectory (about which the model is linearised) shows that there are two distinct





**Figure 9.** Surface origins of NASMW as determined by the backtracking budget analysis (adjoint model simulation) in TS space. Shading indicates the proportion of the budget originating from a particular TS class at the surface within 50 yr. The red box marks the TS definition of NASMW used in this study. Contours show surface density.

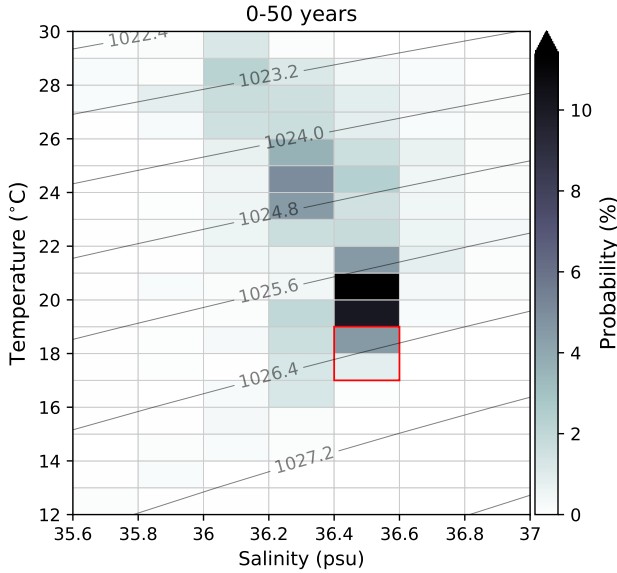

**Figure 10.** Probability distribution of NASMW age (hatched bars) and NASMW age below the mixed layer depth (blue bars). The percentage of the NASMW budget formed in a given 0.5 yr bin is indicated by its associated bar. The expected value of the distribution is marked by a solid line.

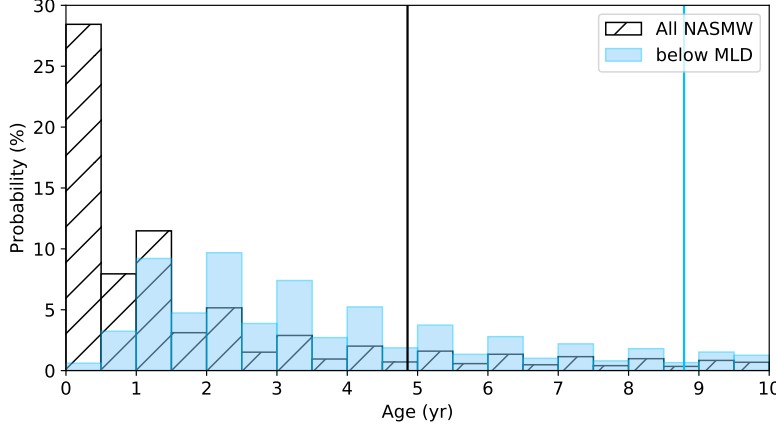

latitudes at which water of this TS signature persistently outcrops (Fig. 2, blue shading). These correspond to the (subpolar) Labrador-Irminger Sea region, southwest of the Greenland-Scotland ridge, and the (Arctic) Greenland Sea region, northeast of





the ridge (Fig. 1, blue shading). The outcrop oscillates between the two regions with the seasonal cycle. The Labrador-Irminger outcrop peaks at the end of March, around a month after the annual local mixed layer depth maximum of $\sim 1$ km. At this peak, water in the NADW TS class occupies $5.3 \times 10^5$ km$^2$ at the surface (Fig. 5), before the area diminishes with the shoaling mixed layer. The Greenland Sea NADW outcrop peaks in November, with a maximal extent of $2.3 \times 10^5$ km$^2$. At this time,

surface NADW is exclusively found northeast of the Greenland-Scotland ridge, and the water mass is shielded from ventilation elsewhere. On average, 66% of the total annual outcrop area is southwest of the ridge, and 34% is to its northeast.

Model NADW volume is almost constant year-round at $4.9 \times 10^7$ km$^3$ (18.7% of the model North Atlantic), deviating by less than 0.5% annually.

## 4.2 Tangent-linear runs

As with NASMW, the dye injection $|c_0\rangle$ coincides with the time step corresponding to the annual maximum outcrop extent, which here falls in April. It should be noted that all of the locations corresponding to this maximum are southwest of the Greenland-Scotland ridge, in the subpolar Labrador-Irminger region. To investigate the nature of the Arctic outcrop, we follow a second injection of dye, in November, when its own outcrop extent is maximal. At this point, the water mass exclusively surfaces within the Arctic circle. We thus refer to these distinct surface waters as SP-NADW (for the southwestern, subpolar

outcrop) and A-NADW (for the northeastern, Arctic outcrop). The tangent-linear model was run for 400 years with the SP-NADW dye injection, and 60 years with the A-NADW dye injection. The run lengths were determined by the rate of surface tracer re-ventilation in each case.

### 4.2.1 SP-NADW

SP-NADW rapidly sinks, with tracer reaching an average depth of 1235 m after 0.2 yr. It initially moves quickly westward,

departing the surface region around Cape Farewell. It then spreads throughout the Labrador basin at depth and extends into the Irminger Sea (Fig. 11) at a temperature of $3.5°$ C (Fig. 12). This initial behaviour is in broad agreement with temperature data from hydrographic sections (e.g. McCartney and Talley, 1982) and spatial distributions captured by CFC measurements (e.g. Rhein et al., 2002), profiling floats (e.g. Bower et al., 2009) and models (e.g. Bower et al., 2011; Gary et al., 2011).

The tracer patch then moves southward, steadily deepening. Its deepest average point, 2466 m, is reached after some 22 yr.

Its mean position is at first closely tied to the DWBC. However, beyond the Flemish Cap, it takes a more central course through the basin interior. While interior southward routes generated by deep eddies have been found parallel to the DWBC in recent profiling float data (Lavender et al., 2000; Bower et al., 2009), our model configuration is non-eddying. As such, these pathways are represented by parameterised turbulent diffusion of the tracer, in a manner which would not be captured by Lagrangian drifters in our model (Gary et al., 2011).

The tracer initialised in the model is quickly sequestered, and is thus not vulnerable to re-exposure. Indeed, while 27% of the initial volume of SP-NADW is re-ventilated within the first decade, only a further 24% is removed during the rest of the century (Fig. 13). The tracer's homogeneity is also preserved, with limited mixing into neighbouring TS classes on decadal time scales (48% remains in the original TS class after 20 years, Fig. 12). There is a tendency over hundreds of years for the





**Figure 11.** As in Fig. 6 but for subpolar-outcropping NADW, and times 2, 10, 20 and 50 yr.

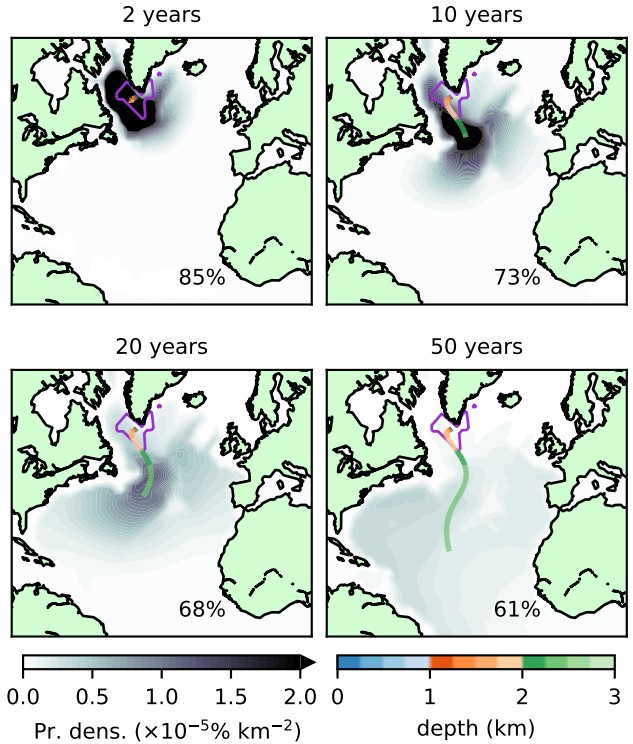

remaining water to cool and freshen to temperatures lower than $1.25°$ C and salinities below $34.65$ psu (Fig. 14). This follows the well-described behaviour of NADW following intrusion into the Southern Ocean. It is known to mix into Circumpolar Deep Water, eventually transforming into bottom water with similar thermohaline characteristics to these (e.g. Orsi et al., 1999).

### 4.2.2  A-NADW

5 The forward evolution of A-NADW (Fig. 15) is altogether different to SP-NADW, categorised by rapid diffusion and depletion. Unlike SP-NADW, which reaches great depths quickly following formation, A-NADW remains close to the surface on the Greenland and Barents shelves, with an average depth of only 502 m after 1 yr. Due to this surface proximity throughout the run, the majority of tracer is re-exposed to the atmosphere. It is accordingly removed by the model's surface restoring scheme, with 95% of the initial volume removed after 20 yr. That which remains either spreads throughout the Arctic ocean, or spills

10 over the shelf to join its SP-NADW counterpart in the Labrador Sea.

  Of all of the initialised A-NADW, that destined for the Atlantic basin represents just 3.8%. We may use the velocity fields of the nonlinear model to estimate transport pathways of this passive tracer into the basin. Consider an idealised case with two



**Figure 12.** As in Fig. 7, but for subpolar-outcropping NADW at 2, 10, 20 and 50 yr. Note that the region enclosed by the red box is the same in both cases.

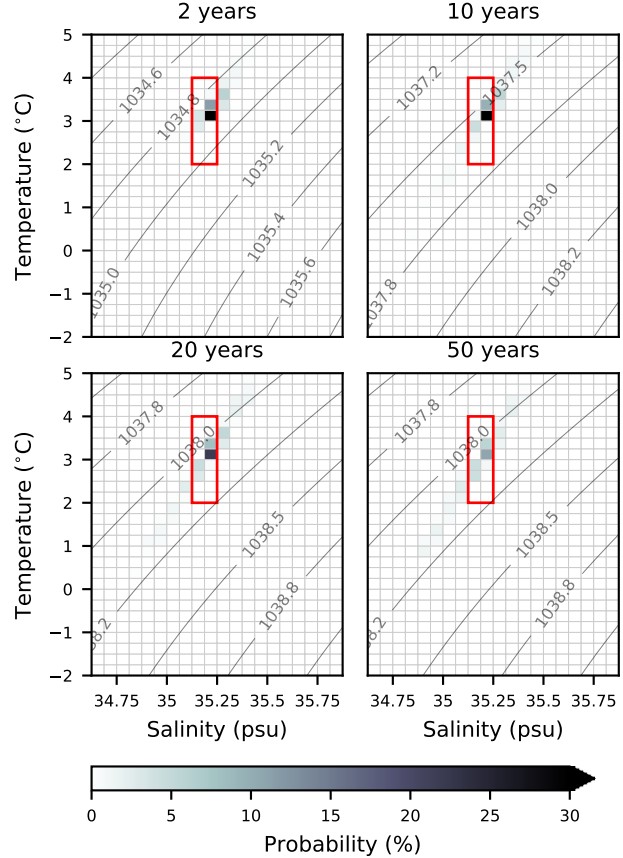

openings into the basin (here taken to represent the Denmark Strait and west of the Reykjanes Ridge), at points $y_1$ and $y_2$. The total volume flux into the basin between two times $t_0$ and $t_1$ will be approximately

$$\Delta V_{t_0,t_1} =$$

$$\int_{t_0}^{t_1} ((u_1(\tau)c_1(\tau)\Delta y_1 \Delta z_1) + (u_2(\tau)c_2(\tau)\Delta y_2 \Delta z_2))d\tau \tag{5}$$

where $c_i$ is the concentration of tracer at opening $i$, $\Delta y_i$ is the width of the opening, $\Delta z_i$ is the depth, $u_i$ is the velocity normal

5  to $\Delta y_i \Delta z_i$, $t_0$ is the injection time and $t_1$ is 50 years later. As all of the above quantities are present in the model output, we may scale this approach up to the model grid and estimate each contribution. In this manner, we find that the vast majority of North-Atlantic-destined A-NADW (88%) enters through the Dennark Strait.

Passive tracer injected at the A-NADW outcrop rapidly moves through TS space (Fig. 16. A colder, fresher water type splinters away from the surface-borne NADW, and as a result, the tracer occupies two distinct regions in TS space for the





**Figure 13.** As in Fig. 6 but for subpolar-outcropping NADW, and times 100, 200 and 400 yr.

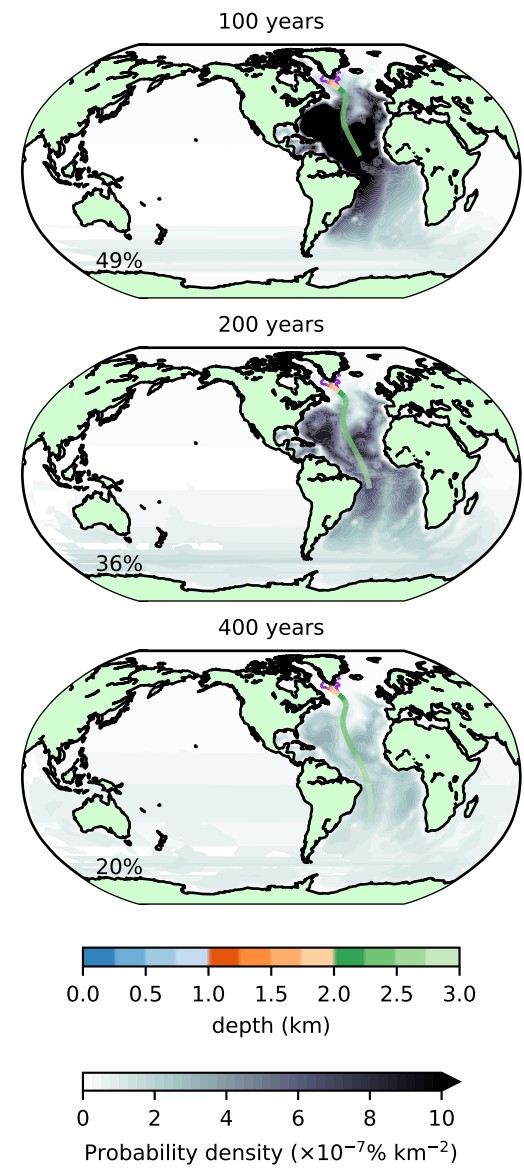

remainder of the run. The warmer of these two water types shares most of its TS properties with those of the NADW at the surface. On the other hand, the colder water type, at $(-0.5, 1)°$ C, bears a similar temperature signature to observed OW (LeBel et al., 2008).





**Figure 14.** As in Fig. 7, but for subpolar-outcropping NADW at 100, 200 and 400 yr.

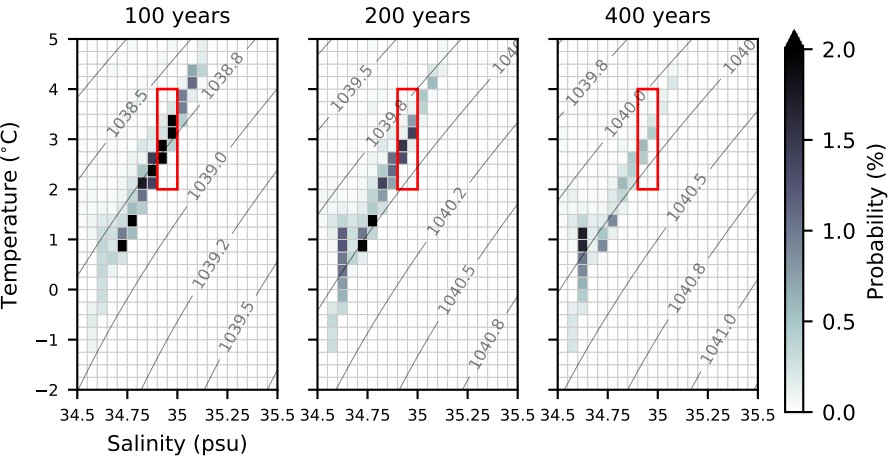

**Figure 15.** As in Fig. 11, but for Arctic-outcropping NADW.

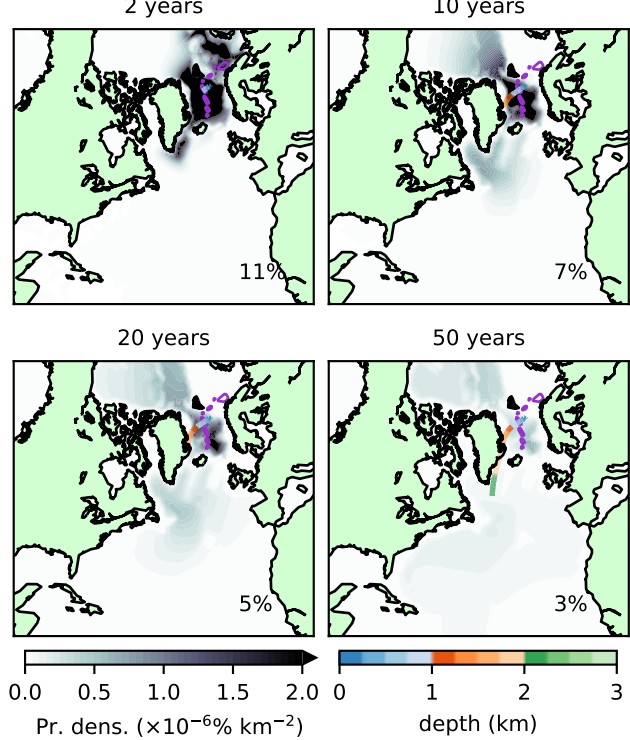

Our simulation shows that surface-borne A-NADW does not proceed to form a substantial part of subsurface NADW. Within 0.6 yr, 78% of the tracer has been re-exposed to the atmosphere. Of that which remains, some 98% has left the NADW TS




**Figure 16.** As in Fig. 12, but for Arctic-outcropping NADW

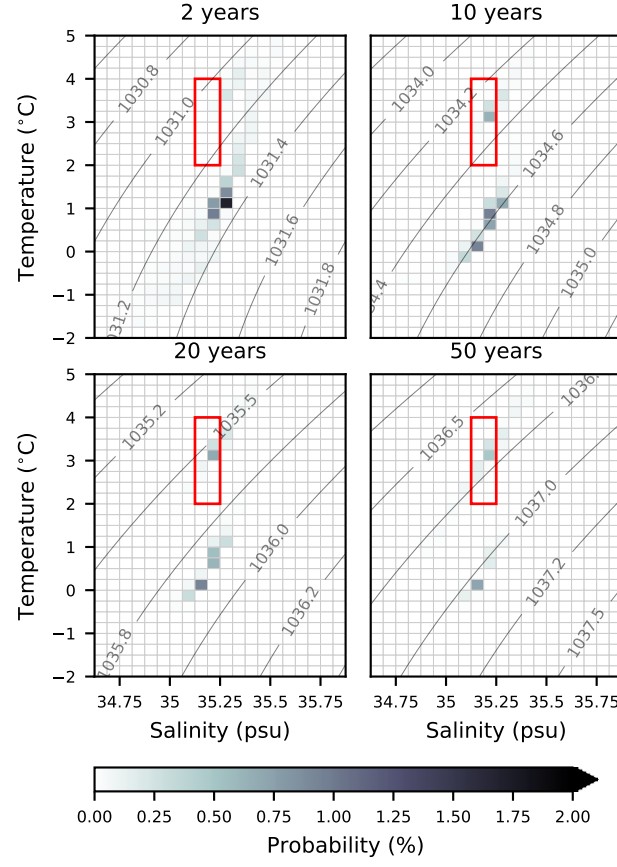

class. The little tracer which stays in the NADW class is persistent, eventually following a similar trajectory to SP-NADW through TS space.

The two tangent-linear experiments (following water from each of the two distinct NADW outcrop regions) suggest that the more northeasterly outcrop contributes quantitatively little to the NADW bulk. Hence, any surface origins of NADW in the Arctic are likely found outside of the NADW TS class.

## 4.3 Adjoint run

As before, we take a water mass budget at the end of the nonlinear model simulation (in this case for NADW after 400 model years) and provide it to the adjoint model. Using this approach, 86% of the NADW budget can be traced to its creation within 400 years (Fig. 17).

During the first year, only 0.13% of the total volume of NADW is tracked back to the surface. The strong presence of shallower NADW during this period leads to a clean signature of the two outcrops (Fig. 17, top panel). On multidecadal time scales, tracer from NADW can be traced back to locations spanning most of the northern North Atlantic (Fig. 17, centre panel),



**Figure 17.** As in Fig. 8, but for NADW and time periods [0,1] yr (top), [0,50] yr (center) and [0,400] yr (bottom)

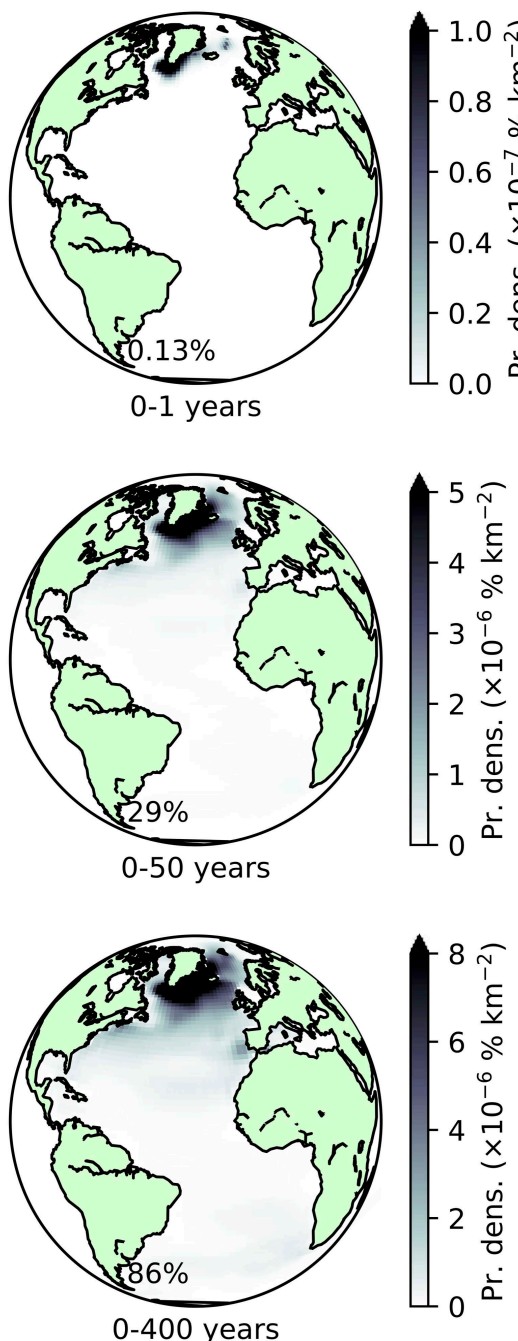





dominated still by the region surrounding its subpolar outcrop. Of particular interest on these time scales is the signature of the Mediterranean Outflow. It has been proposed from hydrographic data that the northward penetration of Mediterranean Water can intermittently reach the subpolar gyre, influencing LSW (Lozier and Stewart, 2008). However, we cannot speculate whether this mechanism is present in our simulation.

NADW spanning the entire 400-year run (Fig. 17, bottom panel) has sources throughout the Atlantic basin, with a notable contribution from the eastern boundary of the South Atlantic, local to the Benguela current. The Labrador-Irminger sector remains the primary source of NADW at all considered time scales, however.

The propagated budget vector can be separated into different source regions and signatures. For instance, 31% of the ventilated NADW can be traced back to the model's Irminger Sea, versus just 14% to the Labrador Sea. We may also construct
a volumetric census of NADW source waters in thermohaline space (Fig. 18, upper left panel). This may further be partitioned into waters originating from the two climatic regions (i.e. subpolar and Arctic) associated with the distinct outcrops of NADW (Fig. 18, upper right and lower left panels, respectively). 45 % of the total NADW formed during the 400-year run may be attributed to the subpolar region. The dominant TS class associated with NADW formation matches that of our definition ($[2,4]°C$, $[34.9, 35.0]$ psu), though there are contributions from a cluster of subpolar water types in the range $[2, 10]°C$,
$[34.5, 35.5]$ psu. Despite the substantial seasonal surface exposure of NADW in the Arctic (34% of the annual outcrop area, see Sect. 4.1), this region ultimately accounts for only 17% of NADW formation in 400 years. This agrees with our findings in the tangent-linear model (Sect. 4.2) that surface-borne NADW in the Arctic is subject to rapid re-ventilation and does not contribute to the NADW bulk. It further suggests that there is no other narrowly defined TS band which outcrops in the Arctic from which Arctic NADW originates. As such, NADW from this region generally derives from a broad range of waters colder
and fresher than NADW.

NADW which does not originate from either of these two regions of the North Atlantic makes up a substantial proportion (24%, Fig. 18, lower right panel) of the budget. Of this, 4% is formed at high latitudes in the Southern hemisphere. 5% of the total originates from outside the Atlantic.

As in Sect. 3, we may also consider the temporal distribution of NADW formation (Fig. 19). Age peaks in the 13th year,
during which 0.6% of the total budget can be traced back to the surface. The expected age is 112 yr. It should be noted that the mean age of model NADW is likely slightly higher, as the results detailed here do not quite account for 100% of the total budget. In Sect. 3.3, we remarked on the unusual formation regions associated with the very oldest North Atlantic Subtropical Mode Water. This was associated with its much shorter life cycle. As such, the corresponding tail of the PDF of NADW age holds fewer surprises. Much like the youngest NADW considered here, the oldest NADW originates predominantly from
winter convection in the Labrador and Irminger seas.

## 5    Discussion and conclusions

We have presented a newly-developed addition to the NEMOTAM tangent-linear and adjoint modelling framework for the NEMO OGCM. This package allows tangent-linear and adjoint tracking of passive-tracer transport. Our framework is rooted



**Figure 18.** As in Fig. 9, but for NADW and time period [0,400] yr. Top left: NADW total volume. Top right: NADW volume traced to the subpolar North Atlantic. Bottom left: NADW volume traced to the Arctic. Bottom right: volume of NADW originating elsewhere.

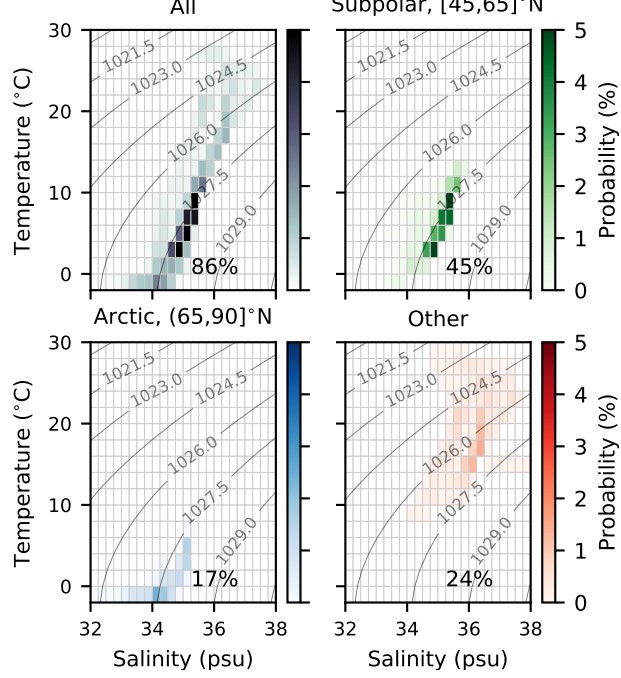

**Figure 19.** As in blue shading of Fig. 10, but for NADW. The black line is the expected age.

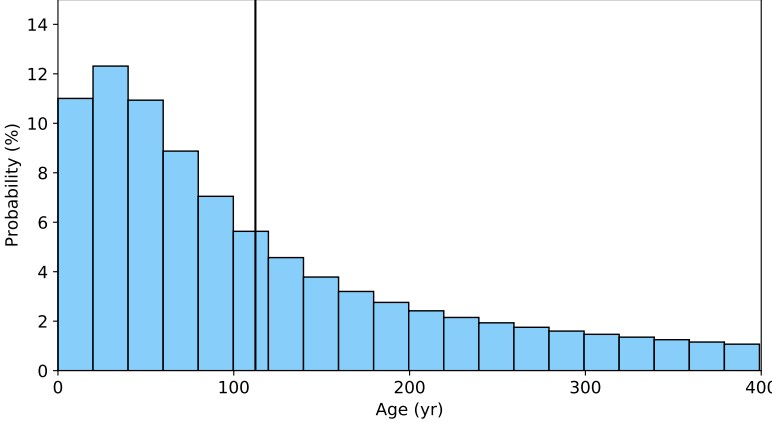

in concentrations and probabilities, comparable to the statistical properties of a high-resolution Lagrangian approach with infinitely many particles. The development was achieved by deactivating dynamic feedbacks in the time-stepping routine of NEMOTAM, and embedding an alternative advection scheme suitable for passive-tracer transport. This advection scheme,





proposed by Fiadeiro and Veronis (1977), is constructed as a linear combination of the upstream and centred schemes, so as to reduce spurious numerical diffusion.

We have exhibited the use of this tool in two case studies concerning the tracking of North Atlantic-borne water masses, North Atlantic Subtropical Mode Water and North Atlantic Deep Water. The method's versatility has been demonstrated
through the calculation of several quantities pertaining to these water masses. If a sufficiently long trajectory is used, a near-complete statistical distribution of surface formation can be constructed. This allows diagnosis of expected age and expected origin location, as well as the rate of eradication and re-ventilation of newly ventilated waters.

The linearity of the method ensures that the water being tracked can be partitioned into several components. When these components are propagated separately, the tracking of the whole is equivalent to the tracking of their union.

The results of our case studies show good agreement between the model configuration and common aspects of prior observational and computational studies. We have shown, for example, that on average, the expected surface origin of tracer initialised within NASMW $32°$ N, $58°$ W (in line with Worthington, 1959; Warren, 1972), that its decay time is around two months (as in Fratantoni et al., 2013), and that a small minority (2.5% in four years) is exported to the subpolar gyre (concurrent with Burkholder and Lozier, 2011). We have estimated the average age of NASMW as 4.5 yr, also close to observations (e.g. Kwon
and Riser, 2004), and shown that a more persistent NASMW subset (with an expected age of over 8.5 yr) underlies the bulk of the water mass (as suggested by Davis et al., 2013).

For NADW, we have shown (in the tangent-linear) that an Arctic outcrop of water with its signature to the northeast of the Greenland-Scotland ridge contributes little to its final form. However, we have found (by backtracking) that Arctic surface waters still make a contribution to NADW formation (17% here), but from a broad range in thermohaline space. It is understood
that OW is produced by the cooling and freshening of North Atlantic inflow to the Greenland Sea (e.g. Quadfasel and Käse, 2007; Dickson et al., 2002). The finer details of this transformation, and subsequent resupply of the transformed water into the North Atlantic are not well-captured by the method. The relative importance of pathways into the Atlantic for shelf water are still poorly known (Macrander et al., 2005), but we find using a broad transport estimate that the Denmark Strait is dominant in the model. For NADW formed in the subpolar outcrop of the Labrador Sea, our tracer spread reflects well that observed using
CFCs (e.g. Rhein et al., 2002). The diffusive behaviour of the passive tracer also means that, despite the non-eddying nature of our model, eddy-driven southward export pathways are better represented than they would be by Lagrangian drifters (e.g. Gary et al., 2011).

Nevertheless, there are several more intricate details of North Atlantic water mass formation and transport which are the subject of ongoing investigation, and as such deserve a dedicated study at higher resolution. For example, the importance of
fine-scale bathymetry for an accurate description of on-shelf NADW formation is well-described (Hansen et al., 2001; LeBel et al., 2008; Smethie Jr and Fine, 2001; Dickson et al., 2002). From a selection of models, Chang et al. (2009) find that the contribution of overflow waters to NADW is misrepresented by models coarser than $\frac{1}{12}°$ in their study. In such models, the Faroe Bank Channel is typically unresolved and the Denmark Strait Overflow resultantly dominates, as is the case here. Furthermore, although our tracer qualitatively reproduces southward export of LSW more effectively than a Lagrangian approach




would, diffusion is still parameterised. It is unknown how well this parameterisation represents sub-grid-scale mixing for such processes.

Despite good broad agreement between the passive-tracer pathways and those noted in previous studies of these water masses, there is an interesting disparity between the forward and backward modes within the TAM itself. This originates from using a TS-based description to inform the initial tracer distribution. For example, the backtracking method suggests that NASMW predominantly originates in slightly warmer surface waters than those of the outcrop used to inform the forward model. Meanwhile, A-NADW, while occupying over a third of the NADW annual outcrop, ultimately contributes almost nothing to the subsurface water mass. These deviations highlight the approximation used by many water-mass-tracking model studies - thermohaline characteristics are not a purely passive tracer. Water parcels experience changes in their thermohaline properties, and so water in a particular TS class at depth is not exclusively related to the same TS class at the surface through a passive advection pathway.

TAM use at high resolution is typically limited, due to baroclinic instability. However, this is not detrimental for passive tracer tracking, due to lack of dynamic feedbacks. As such, our tool may be used in conjunction with higher-resolution configurations of NEMO (e.g. ORCA12, Treguier et al., 2017). The main barrier to higher resolution for users of our development is the necessity of long trajectory outputs, which are required for NEMOTAM operation (Vidard et al., 2015). For the water masses discussed here, we used a 400-year trajectory with repeated forcing. This captured the formation of at least 86% of the water masses considered, leaving a small portion of the budget unaccounted for.

Although we have presented the development in the context of water mass tracking, there are many potential further applications. Ocean heat uptake pathways and carbon sequestration have been studied by means of modelled passive tracers (e.g. Banks and Gregory, 2006; Xie and Vallis, 2012), and adjoint models (e.g. Hill et al., 2004). A slight modification to ignore vertical velocities read from the trajectory would force positive buoyancy on the tracer. This could allow buoyant anthropogenic pollutants to be tracked, with potential application to ocean plastic tracking and oil spill drift prediction. Water mass tracking may itself be complemented by considering continuous (rather than instantaneous) inputs of tracer at the surface. It is hoped that an off-the-shelf tool, bolted onto an existing OGCM will stimulate further research in these areas.

Despite TAM use for sensitivity analysis being traceable to the 1940's (Park and Xu, 2013), its application to ocean science is still in its infancy. The TAM approach is highly versatile, and its application to state-of-the-art OGCMs permits a great many new insights into ocean dynamics. Our development demonstrates the ability of a tangent-linear and adjoint model among the suite of existing water mass tracking methods. It is hoped that this novel tool will encourage new users to realise the potential of this powerful branch of ocean modelling.

*Code availability.* NEMO v3.4 is available from https://forge.ipsl.jussieu.fr/nemo/svn/NEMO/releases/release-3.4 under the CeCILL licence. The configuration presented here is archived on Zenodo for general use (Stephenson, 2019a). The forcing files used in the simulations detailed here are also archived on Zenodo (NEMO Consortium, 2013), as are exact scripts and input files to reproduce our experiments and diagnostics (Stephenson, 2019b)



*Author contributions.* Dafydd Stephenson co-developed the source code, ran simulations and wrote this manuscript. The development of the source code was overseen by Simon Müller, who provided further suggestions and additional technical assistance. Florian Sévellec proposed and supervised the project, and assisted with experiment design. All authors discussed and contributed to the final manuscript.

*Competing interests.* No competing interests are present

5   *Acknowledgements.* This research was supported by the Natural and Environmental Research Council UK (SMURPHS, NE/N005767/1 and MESO-CLIP, NE/K005928/1 and the SPITFIRE DTP) and by the DECLIC and Meso-Var-Clim projects funded through the French CNRS/INSU/LEFE program.





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
