# Peer review of "Tracking water masses using passive-tracer transport in NEMO v3.4 with NEMOTAM: application to North Atlantic Deep Water and North Atlantic Subtropical Mode Water"

_Geoscientific Model Development, 2019_

## Referee Comment (RC1) · Anonymous Referee #1 · 13 Dec 2019

General Comments:

Stephenson et al. describe a promising and interesting addition to the toolbox for visualizing flow in ocean models. This tool is available only within the context of a tangent-linear and adjoint model but it has an advantage over the more commonly used method of Lagrangian particle tracking: insensitivity to the number of particles simulated. The authors then illustrate the types of results possible with their tool with two well known water masses: North Atlantic Subtropical Mode Water and North Atlantic Deep Water. These results are broadly consistent with results from other models and observations.

[Figure]

Finally, as an oceanographer whose expertise is in Lagrangian methods and the water masses discussed in this paper but \*\*not\*\* with tangent-linear and adjoint models (TAM), I found the casting of the TAM approach into a purely passive tracer framework to be exceptionally illuminating; Section 2.2 in particular helped me understand the physicality behind TAMs. In my opinion, this paper contributes both a new method for tracking water masses as well as a special, simplifying case of the TAM approach; both will be useful. I recommend this paper for publication in GMD with some minor clarifications listed below.

Specific Comments:

This paper does not explicitly compare this new method to an established baseline (Lagrangian particle tracking, standard tracer methods). In particular computational demands are cited as one reason for using this method over Lagrangian particles yet there are no run time statistics in the manuscript. In my opinion, explicit quantification of the additional computational resource overhead necessary for running the TAM and this method, and insight into how this overhead scales with model resolution, is necessary in order for other scientists to fully evaluate the merits of this method relative to other methods. In the conclusion, the authors suggest it could be used with higher resolution models (i.e. ORCA12) as an "off-the-shelf" addition. That's definitely intriguing and I think more basic information about resources is necessary to support interest along these lines. Note that here I am \*not\* suggesting that the authors do any particle tracking. Rather, I think it's important to document what kinds of resources are necessary to run this method. Describing these run times will also help illustrate the actual TAM tracer workflow.

Page 2: near line 25: "along with the ability to re-use a single "trajectory" run of a nonlinear model, offer an advantage over passive tracer tracking in a nonlinear OGCM." How exactly is this re-use different from re-using existing output from an eddy-resolving OGCM for different particle tracking studies? (For example, already cited in the manuscript, Gary et al. (2011, 2014) and Burkholder and Lozier (2011) used

different particle tracks from the same model output in three separate papers.)

Page 4, near line 15: The Bower et al. 2009 and 2011 studies did not use profiling floats, only RAFOS floats, which did not profile.

Page 6, near line 10: "Below the surface, it occupies a narrow latitudinal band at depths of up to 240 m (Fig. 2, red shading)." This maximum depth of 240 m is inconsistent with the greater maximum depths reported on page 9 near line 30. Perhaps the 240 m is the time mean maximum depth or a thickness? Please clarify.

Page 6, near line 15, "Additionally, eddy-induced advective velocities, although present in the nonlinear trajectory, were not included for our passive tracers in NEMOTAM." Are these velocities the same as a Gent-McWilliams (GM90) eddy parameterization? If they were used in the model trajectory, why were they not used to move the tracers as well?

Page 12, near line 5, "This suggests that the NASMW surface outcrop is not, in fact, the dominant origin of NASMW...". I am confused by the use of "dominant" in this statement because it seems to me that it is not consistent with Fig. 8a which shows that a significant fraction of the NASMW (up to 70%) comes from the outcrop region. Granted, Fig. 8a is on a 0-5 year time scale and other panels in Fig. 8 clearly show sources from outside the outcrop region on longer time scales, but the contributions of those other sources outside the outcrop region is still less than half of all the NASMW. Please clarify.

---

## Referee Comment (RC2) · Anonymous Referee #2 · 27 Dec 2019

Review of "Tracking water masses using passive-tracer transport in NEMO v3.4 with NEMOTAM: application to North Atlantic Deep Water and North Atlantic Subtropical Mode Water" by Stephenson et al

In this manuscript, the authors present an analysis of the pathways of NADW and NASMW in the NEMOTAM model, focussing on the long time scales and basin-scale pathways in their 2degree resolution simulation.

While the manuscript is well-written and easy to follow, I am not entirely sure it fits the

scope of GMD in its present form. Most of the manuscript is about the physical inter-
pretation of the water mass pathways, and there is relatively little information about the
technical details of the implementation. On the other hand, the low resolution (even for
climate models) means that it is unclear how representative/relevant the results are to
the physical oceanography community. Since there is hardly any comparison to obser-
vational evidence of the pathways (CFCs? C14 dating?), I suspect the manuscript in
its present form would raise questions in a physical oceanography journal too.

I think, however that it should be feasible to rewrite the manuscript to a more traditional
GMD manuscript for the audience here to find it interesting. In particular, I think the
authors would then need to:

1) Add much more information about the technical details of the model. What is its
memory/cpu usage? How does it scale?

2) Add more information about the implementation of the model. What exactly is meant
with a 'perturbation' (page 4, line 28)? A perturbation to what? How does the result
depend on the choice of non-perturbed state? Is that irrelevant because of the as-
sumption of linearity? How good is this assumption of linearity anyways? When and
where does it break down? How big are the resulting errors?

3) Add much more validation of the model implementation. Do the TAM results indeed
agree qualitatively with a full (nonlinear) tracer experiment in the 'normal' model? And
how would this change when changing resolution?

Furthermore, I have the following smaller comments:

- page 1, line 2: Be more specific what is 'probabilistic' about the tool. Is it the diffusive
component?

- page 1, line 5: Be clear that the fact that tracer is removed upon contact with the
surface is a choice?

- page 2, line 1: 'bijective' is a not very common word. Explain?

[Figure]

- page 2, line 8: How many of these floats (order of magnitude) have been deployed?

- page 2, line 15 (and later): it is not true that an 'infinite' number of virtual particles are needed. Depending on machine precision, the tracer concentration is also not simulated to infinite accuracy.

- page 2, line 24: what exactly is meant with a 'probabilitic pathway'?

- page 3, line 9: The recent Bower et al review paper on Atlantic pathways (https://agupubs.onlinelibrary.wiley.com/doi/full/10.1029/2019JC015014) is not mentioned here?

- page 5, line 2: perhaps very briefly explain bra-ket notation to oceanographers?

- page 6, line 4: what is the effect of the non-resolvement of the ice layers in NEMO-TAM? Why are these not implemented? Is it technically impossible?

- page 9, line 23: why not impose an Ertel PV criterion, as is very often done? Is that possible within the TAM framework?

- page 10, line 12: I don't understand the meaning of the word 'mechanically' here

- page 13, line 5: How sure are the authors that this indeed is a lognormal distribution? I would have liked to see a goodness-of-fit analysis. There are other distributions that produce similar-looking PDFs

- Figure 8: I am somewhat surprised that some of the NASMW originates near Greenland? Wasn't one of the conditions that the temperature was higher than 17C? Does that occur within the model in the region near Greenland??

- Figure 10: This bar chart is somewhat confusing because the blue bars raise higher than the striped ones. I assume they are individually normalised? Would it not make more sense to put them on the same y-scale?

- page 16, line 3: There is plenty of evidence that NADW is not formed annually, but

only in some years. That the model does have annual formation clearly is a bias. This should be clearly stated

- page 23, line 26: So where is the rest of the NADW, if it doesn't account for 100%?

- page 25, line 16 (and other places): here, the authors present their model results as if they realistically describe the ocean physics. However, with such a low resolution and known biases, I would prefer to see many more placements of caveats like 'within this model's context' etc

- page 26, line 15: Long runs are not necessary for NEMOTAM to work, but only for the applications chosen here right? If the authors would have focussed on other applications, they could have used shorter runs?

- page 26, line 18: How about 'looping' the NEMO fields? Would that work?

---

## Author Comment (AC1) · 27 Jan 2020

**Authors' response to reviews of the manuscript "Tracking water masses using passive-tracer transport in NEMO v3.4 with NEMOTAM: application to North Atlantic Deep Water and North Atlantic Subtropical Mode Water" by D. Stephenson, S. Müller, and F. Sévellec**

We are grateful to the editor and two referees for their positive and constructive feedback, which has been of great benefit to the work presented in this manuscript. We address their specific comments in detail below.

Font legend:
1. *Reviewers' comments are in black and italic,*
2. Our responses are in black and normal font,
3. Text proposed to be removed from the manuscript is in red and normal font.
4. Text proposed to be added to the manuscript is in blue and normal font.

**Anonymous Referee 1**

**General comments:**

*"Stephenson et al. describe a promising and interesting addition to the toolbox for visualizing flow in ocean models. This tool is available only within the context of a tangentlinear and adjoint model but it has an advantage over the more commonly used method of Lagrangian particle tracking: insensitivity to the number of particles simulated. The authors then illustrate the types of results possible with their tool with two well known water masses: North Atlantic Subtropical Mode Water and North Atlantic Deep Water. These results are broadly consistent with results from other models and observations. Finally, as an oceanographer whose expertise is in Lagrangian methods and the water masses discussed in this paper but \*\*not\*\* with tangent-linear and adjoint models (TAM), I found the casting of the TAM approach into a purely passive tracer framework to be exceptionally illuminating; Section 2.2 in particular helped me understand the physicality behind TAMs. In my opinion, this paper contributes both a new method for tracking water masses as well as a special, simplifying case of the TAM approach; both will be useful. I recommend this paper for publication in GMD with some minor clarifications listed below."*

We thank the referee for this positive and encouraging response. We are pleased to find that our treatment of the methodology was insightful to a modeller outside of the versatile but admittedly niche field of tangent-linear and adjoint modelling.

**Specific comments**

1. *"This paper does not explicitly compare this new method to an established baseline (Lagrangian particle tracking, standard tracer methods). In particular computational demands are cited as one reason for using this method over Lagrangian particles yet there are no run time statistics in the manuscript. In my opinion, explicit quantification of the additional computational resource overhead necessary for running the TAM and this method, and insight into how this overhead scales with model resolution, is necessary in order for other scientists to fully evaluate the merits of this method relative to other methods. In the conclusion, the authors suggest it could be used with higher resolution models (i.e. ORCA12) as an "off-the-shelf" addition. That's definitely intriguing and I think more basic information about resources is necessary to support interest along these lines. Note that here I am \*not\* suggesting that the authors do any particle tracking. Rather, I think it's important to document what kinds of resources are necessary to run this method. Describing these run times will also help illustrate the actual TAM tracer workflow"*

A variation on this comment regarding runtime statistics was also brought to our attention by the second reviewer, and is an important point which we had missed. Thank you to both reviewers for the recommendation, which we have now addressed using a number of efficiency experiments at laminar and eddying resolutions, detailed below in a proposed additional section:

**2.4 Performance**

In order to test performance, the model was run for one-hundred days at 2º resolution (the ORCA2 configuration used in our case studies) and five days at ¼º resolution (the ORCA025 configuration). Each configuration was run in four different modes (nonlinear, nonlinear with TAM trajectory production, tangent-linear with passive tracer, adjoint with passive tracer). Trajectory files were written once per day, and the linear advection scheme was the weighted-mean scheme described in Section 2.3. Each test was conducted with a range of parallelisation arrangements (16, 32, 64, 128, 256 and 512 CPU cores) and repeated 10 times to account for system variability (Table. 1). The tests were conducted on a local HPC system, with 72 compute nodes, each offering two eight-core 2.6GHz processors (Intel Xeon E5-2650 v2) and 64GiB, connected by an InfiniBand QDR network.

*Table 1. Results of performance tests for two model configurations (ORCA2, upper section and ORCA025, lower section). Top rows: trajectory storage requirements per output. Lower rows: runtime per model day for four model modes: Nonlinear with trajectory output (NL[T]), nonlinear without trajectory output (NL), tangent-linear (TL) and adjoint (AD). The standard deviation of ten runs is given as a ± uncertainty.*

| Cores (nodes) | | | 16 (1) | 32 (2) | 64 (4) | 128 (8) | 256 (16) | 512 (32) |
|---|---|---|---|---|---|---|---|---|
| ORCA2 | | | | | | | | |
| Trajectory size (K/d) | | | 43909.15 | 44417.76 | 43977.98 | 44412.36 | 44847.80 | 46351.24 |
| Mode | NL [T] | Runtime (s/d) | 2.09 ±0.20 | 1.54 ±0.17 | 1.23 ±0.17 | 1.39 ±0.26 | 2.53 ±0.27 | 4.08 ±0.35 |
| | NL | | 1.98 ±0.17 | 1.372 ±0.10 | 1.10 ±0.09 | 0.99 ±0.08 | 1.85 ±0.36 | 2.18 ±0.70 |
| | TL | | 0.78 ±0.01 | 0.50 ±0.01 | 0.36 ±0.09 | 0.36 ±0.13 | 0.46 ±0.18 | 0.88 ±0.10 |
| | AD | | 0.89 ±0.01 | 0.61 ±0.02 | 0.53 ±0.04 | 0.50 ±0.11 | 0.59 ±0.05 | 1.99 ±0.61 |
| ORCA025 | | | | | | | | |
| Trajectory size (K/d) | | | | 5217427.33 | 5218494.66 | 5220824.00 | 5217658.00 | 5218448.00 |
| Mode | NL [T] | Runtime (s/d) | Insufficient memory | 1091.55 ±69.38 | 692.98 ±21.38 | 428.36 ±13.76 | 302.18 ±30.55 | 143.6 ±12.73 |
| | NL | | | 1065.48 ±54.56 | 656.94 ±13.74 | 386.34 ±13.89 | 262.72 ±34.56 | 131.7 ±13.28 |
| | TL | | Insufficient memory | | 315.26 ±26.17 | 207.82 ±14.43 | 136.42 ±13.00 | 64.7 ±4.20 |
| | AD | | | 403.74 ±36.18 | 252.36 ±14.17 | 181.12 ±7.70 | 92.4 ±6.26 |

The general order of time efficiency is consistent across all tests: the linear forward model is between 2.5 (16 cores) and 3.1 (64 cores) times as fast as the nonlinear model without trajectory output in ORCA2, and 1.9 (64 cores) to 2.0 (512 cores) times as fastre in ORCA025. The adjoint is slightly less efficient. The nonlinear runs which produce trajectory output are the slowest across the board, due to the high level of output. Memory use appears higher in the linear model than the nonlinear, such that the linear model could not be run on two 64GiB compute nodes alone in ORCA025 (column two).

The model shows good scaling in the ORCA025 configuration at all CPU arrangements tested here, but begins to worsen in ORCA2 beyond 128 CPU cores. Further, the added time required for trajectory output in nonlinear ORCA2 runs on 128 CPU cores can be considerable. This is possibly due to the generation of a very large number of files during the model run, as the number of files generated is proportional to the number of CPU cores, the trajectory write frequency, and the run length. The required storage size remains relatively constant for runs with larger number of CPU cores, despite the greater number of files (Table 1, first row).

An additional limitation which we discovered during our longest experiments comes from system file-number limits, which can readily be reached for typical systems for very long runs using many cores.

2. *Page 2: near line 25: "along with the ability to re-use a single "trajectory" run of a nonlinear model, offer an advantage over passive tracer tracking in a nonlinear OGCM." How exactly is this re-use different from re-using existing output from an eddy-resolving OGCM for different particle tracking studies? (For example, already cited in the manuscript, Gary et al. (2011, 2014) and Burkholder and Lozier (2011) used different particle tracks from the same model output in three separate papers.)*

We agree with the reviewer that similar offline methods such as Lagrangian particle tracking can indeed re-use identical model outputs for distinct particle release experiments. Our comment is perhaps unclearly worded (having followed a comment regarding Lagrangian methods) but refers instead to online passive tracer methods.

This native ability to track water both forward and backward in time, along with the ability to re-use a single "trajectory" run of a nonlinear model, offer an advantage over passive tracer tracking in a nonlinear OGCM
>
While the passive tracer framework offers an advantage over the Lagrangian technique through its probabilistic perspective, the *linear* passive tracer method also offers an advantage over the tracking of an online passive tracer in a nonlinear OGCM. In particular, it natively allows the user to track water both forward and backward in time, while re-using the output of a single run.

3. *Page 4, near line 15: The Bower et al. 2009 and 2011 studies did not use profiling floats, only RAFOS floats, which did not profile*

Thank you to the referee for bringing this to our attention.
However, new data from profiling floats and modelled Lagrangian drifters (Bower et al., 2009, 2011) suggest that this is not the complete story.
>
However, new data from floats and modelled Lagrangian drifters (Bower et al., 2009, 2011) suggest that this is not the complete story.

4. *Page 6, near line 10: "Below the surface, it occupies a narrow latitudinal band at depths of up to 240 m (Fig. 2, red shading)." This maximum depth of 240 m is inconsistent with the greater maximum depths reported on page 9 near line 30. Perhaps the 240 m is the time mean maximum depth or a thickness? Please clarify.*

Thanks to the reviewer for highlighting this inconsistency. The value quoted does indeed refer to thickness. The correct value is in fact 310m, and has been corrected below. This also revealed an error in the y-axis tick labels of Figure 2, which has now also been corrected.

"Below the surface, it occupies a narrow latitudinal band at depths of up to 240 m (Fig. 2, red shading)."
>
"Below the surface, it occupies a narrow latitudinal band at a maximum depth of 310 m (Fig. 2, red shading)."

5. *Page 6, near line 15, "Additionally, eddy-induced advective velocities, although present in the nonlinear trajectory, were not included for our passive tracers in NEMOTAM." Are these velocities the same as a Gent-McWilliams (GM90) eddy parameterization? If they were used in the model trajectory, why were they not used to move the tracers as well?*

As the referee supposes, these are the GM90 bolus velocities. They are by default in NEMOTAM 3.4 not used for tracer advection. In particular, the nonlinear model calculates them online, and does not explicitly store them in the trajectory, so that they are not seen by NEMOTAM. Instead, the iso-neutral slopes used for their computation are stored, (so EIV can, in fact, be retrieved). We have since, for a different adjoint study, enabled online EIV calculation using these slopes in NEMOTAM by calling the corresponding routine from

the nonlinear model. This has allowed us to test the impact on passive-tracer evolution. Preliminary tests reveal little difference for the case studies described in the manuscript (Fig A,B,C,D,E), but we have added the option of EIV to the source code for further studies.

[Figure]

*Fig A: Remaining volume of tangent-linear experiments additionally showing normalised absolute upper limit of error. Dark, dashed lines show the volume of tracer remaining without EIV. Dark solid lines are the equivalent with EIV. Faint lines show worst-case limits: EIV-on volume, +/- the globally integrated absolute volume error in every grid cell. All curves are normalised by the injected tracer volume. (Note that blue lines overlap, because of the small sensitivity of this experiment to EIV.)*

[Figure]

*Fig B: Snapshots of depth-integrated volume for tangent-linear NASMW propagation without (top row) and with (middle row) eddy-induced velocities at 1y (left column) 5y (centre column) and 10y (right column). The bottom row shows the difference between the two. (Note the smaller colorbar extent of the last column, demonstrating that the difference is roughly one order of magnitude smaller than the results.)*

[Figure]

*Fig C: As in Figure B, but for subpolar-outcropping NADW. (Note smaller colorbar extent of the last column, demonstrating that the difference is two orders of magnitude as small as the results.)*

[Figure]

*Fig. D: As in Fig. C, but for Arctic-outcropping NADW. (Note smaller colorbar extent of the last column, demonstrating that the difference is two orders of magnitude as small as the results.)*

[Figure]

*Fig E: Ventilated tracer volume of NASMW (left) and NADW (right) over the first ten years of case study experiments, with (top row) and without (middle row) eddy-induced velocities. The bottom row shows the difference. (Note smaller colorbar extent of the last column, demonstrating that the difference is two orders of magnitude as small as the results.)*

We suggest the following edit to the manuscript:

Additionally, eddy-induced advective velocities, although present in the nonlinear trajectory, were not included for our passive tracers in NEMOTAM.
>
Additionally, eddy-induced advective velocities are computed online by the nonlinear model and not stored explicitly in the trajectory. They may be recomputed from other trajectory variables, but this is not the default behaviour of NEMOTAM. As such, they were not included for the passive-tracer experiments in this study. They have been since been enabled in the passive-tracer source code as an optional online calculation. A preliminary comparative test suggests that errors are minor for our case studies (~6% in the worst case), but we expect the difference to be greater in steep isopycnal regions such as the Southern Ocean.

6. *Page 12, near line 5, "This suggests that the NASMW surface outcrop is not, in fact, the dominant origin of NASMW...". I am confused by the use of "dominant" in this statement because it seems to me that it is not consistent with Fig. 8a which shows that a significant fraction of the NASMW (up to 70%) comes from the outcrop region. Granted, Fig. 8a is on a 0-5 year time scale and other panels in Fig. 8 clearly show sources from outside the outcrop region on longer time scales, but the contributions of those other sources outside the outcrop region is still less than half of all the NASMW. Please clarify*

We thank the referee for highlighting this. We agree that Fig. 8a gives the impression that the majority of NASMW comes from the outcrop region, and have now clarified this inside the text. Unlike in the forward model, where spatial maps represent snapshots, the adjoint requires time-integrated fields be shown. As such we did not include a single outcrop contour for this figure as we thought that this would be misleading given the strong seasonal behaviour of the outcrop (it is non-existent for half of the year). It is thus difficult to convey whether or not tracer ventilated within or outside of the outcrop with this figure, for which a TS-histogram is more appropriate (Fig. 9). In our TS budget, it is more clear that the dominant origin indeed has a warmer signature than surface water within the outcrop.

Inset percentages show the global integral (that is, the total proportion of the budget which is formed during each time period).
>
Inset percentages show the global integral (that is, the total proportion of the budget which is formed during each time period). Note that, contrary to Fig. 6, which displayed instantaneous fields, here time-integrated fields are displayed. Due to the large variability of its extent over the integration window, the outcrop region is not shown.
(Fig. 8, caption)

This suggests that the NASMW surface outcrop is not, in fact(…)
>
While the surface distribution suggests that a relatively small neighbourhood dominates the formation of model NASMW (Fig. 8), we remind that the seasonal variability of surface properties (particularly the outcrop area) reflects strongly on the thermohaline properties of NASMW at formation (Fig. 9). This suggests that the NASMW surface outcrop is not, in fact(…)

**Anonymous Referee 2**

**General comments**

> *"In this manuscript, the authors present an analysis of the pathways of NADW and NASMW in the NEMOTAM model, focussing on the long time scales and basin-scale pathways in their 2degree resolution simulation. While the manuscript is well-written and easy to follow, I am not entirely sure it fits the scope of GMD in its present form. Most of the manuscript is about the physical interpretation of the water mass pathways, and there is relatively little information about the technical details of the implementation. On the other hand, the low resolution (even for climate models) means that it is unclear how representative/relevant the results are to the physical oceanography community. Since there is hardly any comparison to observational evidence of the pathways (CFCs? C14 dating?), I suspect the manuscript in its present form would raise questions in a physical oceanography journal too.*
>
> *I think, however that it should be feasible to rewrite the manuscript to a more traditional GMD manuscript for the audience here to find it interesting. In particular, I think the authors would then need to:"*

We thank the reviewer for their suggestions, which we address individually below. We remark that our decision to submit this manuscript to GMD was based on perceived benefit to the community of the tool (especially given the existence of the NEMO special issue), rather than to establish new ground in the analysis of either of the considered water masses. These case studies were chosen to showcase the versatility of the technique, and highlight the possibilities available to a scientist undertaking a dedicated study at higher resolution.

> 1. *Add much more information about the technical details of the model. What is its memory/cpu usage? How does it scale?*

We are glad that this has been brought to our attention by both reviewers and express our gratitude again for highlighting this oversight. We have prepared an additional section to be added to the manuscript:

**2.4 Performance**

In order to test performance, the model was run for one-hundred days at 2° resolution (the ORCA2 configuration used in our case studies) and five days at ¼° resolution (the ORCA025 configuration). Each configuration was run in four different modes (nonlinear, nonlinear with TAM trajectory production, tangent-linear with passive tracer, adjoint with passive tracer). Trajectory files were written once per day, and the linear advection scheme was the weighted-mean scheme described in Section 2.3. Each test was conducted with a range of parallelisation arrangements (16, 32, 64, 128, 256 and 512 CPU cores) and repeated 10 times to account for system variability (Table. 1). The tests were conducted on a local HPC system, with 72 compute nodes, each offering two eight-core 2.6GHz processors (Intel Xeon E5-2650 v2) and 64GiB, connected by an InfiniBand QDR network.

*Table 1. Results of performance tests for two model configurations (ORCA2, upper section and ORCA025, lower section). Top rows: trajectory storage requirements per output. Lower rows: runtime per model day for four model modes: Nonlinear with trajectory output (NL[T]), nonlinear without trajectory output (NL), tangent-linear (TL) and adjoint (AD). The standard deviation of ten runs is given as a ± uncertainty.*

| Cores (nodes) | 16 (1) | 32 (2) | 64 (4) | 128 (8) | 256 (16) | 512 (32) |
|---|---|---|---|---|---|---|
| ORCA2 | | | | | | |
| Trajectory size (K/d) | 43909.15 | 44417.76 | 43977.98 | 44412.36 | 44847.80 | 46351.24 |

| Mode | Runtime (s/d) | | | | | | |
|---|---|---|---|---|---|---|---|
| NL [T] | | 2.09 ±0.20 | 1.54 ±0.17 | 1.23 ±0.17 | 1.39 ±0.26 | 2.53 ±0.27 | 4.08 ±0.35 |
| NL | | 1.98 ±0.17 | 1.372 ±0.10 | 1.10 ±0.09 | 0.99 ±0.08 | 1.85 ±0.36 | 2.18 ±0.70 |
| TL | | 0.78 ±0.01 | 0.50 ±0.01 | 0.36 ±0.09 | 0.36 ±0.13 | 0.46 ±0.18 | 0.88 ±0.10 |
| AD | | 0.89 ±0.01 | 0.61 ±0.02 | 0.53 ±0.04 | 0.50 ±0.11 | 0.59 ±0.05 | 1.99 ±0.61 |
| ORCA025 | | | | | | | |
| Trajectory size (K/d) | | | 5217427.33 | 5218494.66 | 5220824.00 | 5217658.00 | 5218448.00 |
| NL [T] | Runtime (s/d) | Insufficient memory | 1091.55 ±69.38 | 692.98 ±21.38 | 428.36 ±13.76 | 302.18 ±30.55 | 143.6 ±12.73 |
| NL | | | 1065.48 ±54.56 | 656.94 ±13.74 | 386.34 ±13.89 | 262.72 ±34.56 | 131.7 ±13.28 |
| TL | | Insufficient memory | | 315.26 ±26.17 | 207.82 ±14.43 | 136.42 ±13.00 | 64.7 ±4.20 |
| AD | | | | 403.74 ±36.18 | 252.36 ±14.17 | 181.12 ±7.70 | 92.4 ±6.26 |

The general order of time efficiency is consistent across all tests: the linear forward model is between 2.5 (16 cores) and 3.1 (64 cores) times as fast as the nonlinear model without trajectory output in ORCA2, and 1.9 (64 cores) to 2.0 (512 cores) times as fast in ORCA025. The adjoint is slightly less efficient. The nonlinear runs which produce trajectory output are the slowest across the board, due to the high level of output. Memory use appears higher in the linear model than the nonlinear, such that the linear model could not be run on two 64GiB compute nodes alone in ORCA025 (column two).

The model shows good scaling in the ORCA025 configuration at all CPU arrangements tested here, but begins to worsen in ORCA2 beyond 128 CPU cores. Further, the added time required for trajectory output in nonlinear ORCA2 runs on 128 CPU cores can be considerable. This is possibly due to the generation of a very large number of files during the model run, as the number of files generated is proportional to the number of CPU cores, the trajectory write frequency, and the run length. The required storage size remains relatively constant for runs with larger number of CPU cores, despite the greater number of files (Table 1, first row).
An additional limitation which we discovered during our longest experiments comes from system file-number limits, which can readily be reached for typical systems for very long runs using many cores.

2. *Add more information about the implementation of the model. What exactly is meant with a 'perturbation' (page 4, line 28)? A perturbation to what? How does the result depend on the choice of non-perturbed state? Is that irrelevant because of the assumption of linearity? How good is this assumption of linearity anyways? When and where does it break down? How big are the resulting errors?*

It was not our intention to provide an in-depth description of the TAM framework in a general context outside of our application. The passive tracer approach is a simplification of an otherwise quite technical method (TAM) for which there is an extensive literature. Our hope was to focus on this simplification motivated by its potential for accessibility and interdisciplinary use. We have now clarified this from the outset, and thank the reviewer for bringing this to our attention.

These relationships are derived in full by Errico (1997).
>
These relationships are derived in full by Errico (1997), who provides a thorough treatment of the linear method and its limitations, beyond the simplified, inherently linear use-case of the present study.

Regarding the dependency on the non-perturbed state (the trajectory), in the passive tracer context the perturbation (or as we prefer, "injection") does not actually change anything, and so can be seen as an analysis of the trajectory. In this sense, it is entirely dependent on it.

The assumption of linearity and window of validity for a TAM is an example of a typical complication (c.f. Errico et al., 1993, Tellus), but one which is not an issue in the passive tracer case, as there are no feedbacks. For instance, static instability induced by a decrease in temperature, due to the high degree of nonlinearity, would not be triggered by an active linear perturbation, which is an error of the active linear model. However, in a passive linear model, no response is ever triggered by the tracer, and so these errors cannot arise. As these problems are not present in the simplified case of passive tracer tracking, we choose not to describe them, as they lie outside of the realm of our study.

3. Add much more validation of the model implementation. Do the TAM results indeed agree qualitatively with a full (nonlinear) tracer experiment in the 'normal' model? And how would this change when changing resolution?

This is an interesting question in that, despite there being no effective difference between a linear and a nonlinear passive tracer, there remain some important simplifications in the TAM. In particular, the advection scheme is linearised (such that the adjoint is consistent). The nonlinear advection scheme (TVD) is considered in Section 2.3. As described (page 6, line 15-20) some additional parameterisations are also absent in the linear advection-diffusion routines, particularly EIV. This is the native behaviour of NEMOTAM, but we have since changed it for another adjoint study, allowing us to conduct some tests on passive-tracer propagation. Preliminary tests reveal little difference for the case studies described in the manuscript, but we have added EIV as a switch in the source code for subsequent studies. (Note that we anticipate that in specific regions such as the Antarctic Circumpolar Current, which is much more intense in terms of mesoscale eddy turbulence, this will make a larger difference.) Regarding this, we have suggest a modification to the manuscript to discuss that limitation:

Additionally, eddy-induced advective velocities, although present in the nonlinear trajectory, were not included for our passive tracers in NEMOTAM.
>
Additionally, eddy-induced advective velocities are computed online by the nonlinear model and not stored explicitly in the trajectory. They may be recomputed from other trajectory variables, but this is not the default behaviour of NEMOTAM. As such, they were not included for the passive-tracer experiments in this study. They have since been enabled in the passive-tracer source code as an optional online calculation. A preliminary comparative test suggests that errors are minor for our case studies (~6% in the worst case), but we expect the difference to be greater in steep isopycnal regions such as the Southern Ocean.

**Specific comments**
1. *page 1, line 2: Be more specific what is 'probabilistic' about the tool. Is it the diffusive component?*

This question stimulated an interesting discussion for which we thank the reviewer. In the broadest sense, the Eulerian frame of reference is what determines the probabilistic nature of the approach. Parameterised diffusion, offering sub-grid-scale closure, is an important component of this, but does not determine the probabilistic nature of the approach alone. Rather, in terms of the equations themselves, the advection and diffusion of a concentration in the Eulerian framework rather than of a particle in the Lagrangian perspective, is what allows us to produce a continuous probability distribution natively. This is in opposition to reconstructing one from a series of discrete trajectories, as in the Lagrangian framework.

While we choose to maintain the current wording in the abstract for brevity, we propose to elaborate further in the Introduction (page 2, line 20-24):

However, due to its probablistic nature, it offers an advantage over the more conventional tracking technique of Lagrangian particle modelling. Within the Lagrangian framework, ocean sensitivity to initial conditions means that a very large ensemble of initially close particle deployments may be required to representatively sample the full space of particle trajectories. A TAM can be exploited to bypass this requirement, producing a continuous probability distribution of all possible particle trajectories. As such, the method does not describe

particle locations and deterministic trajectories, as a Lagrangian approach would, but tracer concentrations and probabilitic pathways.
>
However, in the passive-tracer case it offers an advantage over the more conventional tracking technique of Lagrangian particle modelling, in that it is inherently probabilistic. The Eulerian framework allows for the modelled propagation of a continuous field (tracer concentration), corresponding to a continuous probability distribution. (In the Lagrangian perspective, an estimate of such a distribution would have to be reconstructed from a large number of discrete trajectories.) Hence, whereas the Lagrangian approach describes particle locations and deterministic trajectories, the Eulerian framework describes tracer concentrations and probabilistic pathways.

> 2. *page 1, line 5: Be clear that the fact that tracer is removed upon contact with the surface is a choice?*

Thanks to the referee. This was previously unclear and has been rewritten.

Upon contact with the surface, the tracer is removed from the system, and a record of ventilation is produced.
>
To represent surface (re-)ventilation, we optionally decrease the tracer concentration in the surface layer, and record this concentration removal to produce a ventilation record.

> 3. *page 2, line 1: 'bijective' is a not very common word. Explain?*

While we agree that the word is perhaps uncommon in oceanography, it has a strict mathematical definition (a one-to-one relationship) which is an important aspect of the early mathematical diagnoses of subduction described in this line. As such, we choose to maintain its use.

> 4. *page 2, line 8: How many of these floats (order of magnitude) have been deployed?*

The sentence could be modified as follows for clarity:

Despite the number of these floats, these pathways remain under-sampled > Despite their number (typically several tens of floats), these pathways remain under-sampled.

> 5. *page 2, line 15 (and later): it is not true that an 'infinite' number of virtual particles are needed. Depending on machine precision, the tracer concentration is also not simulated to infinite accuracy.*

We thank the referee for this important technical point, which we had overlooked. Our intention was to highlight the continuous vs. discrete nature of tracer vs. particle tracking. The manuscript has been revised accordingly:

As such, an infinite number of them is required to fully represent an equivalent dye concentration. > As such, a theoretically infinite number of them is required to fully represent an equivalent dye concentration (although in practice a large enough finite number can allow statistical convergence).

Our framework is rooted in concentrations and probabilities, comparable to the statistical properties of a high-resolution Lagrangian approach with infinitely many particles. > Our framework is rooted in concentrations and probabilities, comparable to the statistical properties of a Lagrangian approach with a large (theoretically infinite) number of particles.

> 6. *page 2, line 24: what exactly is meant with a 'probabilitic pathway'?*

This line was modified as part of a suggested revision in response to comment 1.

As such, the method does not describe particle locations and deterministic trajectories, as a Lagrangian approach would, but tracer concentrations and probabilitic pathways. > Hence, whereas Lagrangian approach describes particle locations and deterministic trajectories, Lagrangian approach describes tracer concentrations and probabilistic pathways.

7. *page 3, line 9: The recent Bower et al review paper on Atlantic pathways (https://agupubs.onlinelibrary.wiley.com/doi/full/10.1029/2019JC015014) is not mentioned here?*

While this paper is referenced elsewhere (Page 2, line 8), it describes Lagrangian pathways rather than attempts to linearly model passive tracers, which is the focus of this line.

8. *page 5, line 2: perhaps very briefly explain bra-ket notation to oceanographers? –*
We agree that the notation is uncommon, and suggest adding the following short explanation:

The linear evolution of an initial perturbation |**u**> to the ocean state vector is described by the equation
lu(t)> =          Psi(t, t0)|**u**0>  (1)
 where lu0> provides the condition of the perturbation at time t0, Psi(t, t0) is known as the propagator matrix and we have used the "bra-ket" notation of Dirac (1939).
>
The linear evolution of an initial perturbation |**u**> to the ocean state vector is described by the equation
lu(t)> =          Psi(t, t0)|**u**0>  (1)
 where lu0> provides the condition of the perturbation at time t0 and Psi(t, t0) is known as the propagator matrix. We have used the "bra-ket" notation of Dirac (1939) in which row and column vectors are respectively written as bras (<al) and kets (lb>) such that closed bra-ket terms become scalar (through the scalar product : <alb> = c).

9. *page 6, line 4: what is the effect of the non-resolvement of the ice layers in NEMOTAM? Why are these not implemented? Is it technically impossible?*

This is a well-raised point which requires further elaboration. In the context of an active tracer, ice formation and melt is highly nonlinear, and represents an error in NEMOTAM's linearization approach. However, a passive tracer would not interact with these processes. Furthermore, any change in volume is handled in the trajectory, so concentration changes due to, for example, dilution, are pre-emptively addressed by the trajectory.

There are two ice layers (Fichefet and Maqueda, 1997) in the background state, although these are not incorporated into NEMOTAM.
>
There are two ice layers (Fichefet and Maqueda, 1997) in the background state. The ice model is not linearised or coupled to NEMOTAM, although in the context of this study, ice dynamics are inherently unaffected (as a tracer which is passive cannot affect ice).

10. *page 9, line 23: why not impose an Ertel PV criterion, as is very often done? Is that possible within the TAM framework?*

We had considered several criteria when preparing the experiments, and PV would have indeed been possible (any property of the background state can be used to determine where to inject the tracer). Given the lack of a universally accepted NASMW definition and in the interests of simplicity, we used a thickness criterion, as in Gary et al. (2014, J. Phys. Ocean.).

Here, we define NASMW as > While the method allows us to define water masses in terms of any model variable, we choose for simplicity to utilise the common approach of a temperature-salinity range. In particular, we define NASMW as

11. *page 10, line 12: I don't understand the meaning of the word 'mechanically' here*

This appears to be a typo and has been corrected. We thank the referee for bringing this to our attention.

As time progresses, because of vertical mixing together with near-surface tracer removal by the restoring scheme, remaining tracer mechanically tends to reach deeper, colder, fresher waters.
>
As time progresses, because of vertical mixing together with near-surface tracer removal by the restoring scheme, remaining tracer tends to reach deeper, colder, fresher waters.

12. *page 13, line 5: How sure are the authors that this indeed is a lognormal distribution? I would have liked to see a goodness-of-fit analysis. There are other distributions that produce similar-looking PDFs*

We agree with the referee that this description is too specific and suggest its removal:

This distribution resembles a log-normal PDF with a lower skewness for NASMW lying below the MLD (Fig. 10, blue bars).
>
There is a visibly lower skewness in the PDF of NASMW lying below the MLD (Fig. 10, blue bars).

13. *Figure 8: I am somewhat surprised that some of the NASMW originates near Greenland? Wasn't one of the conditions that the temperature was higher than 17C? Does that occur within the model in the region near Greenland??*

After injection, the tracer is not tied to any particular temperature range, or indeed water mass. Under the assumption that the pathway of the tracer-tagged water is representative of the history of NASMW, it must be concluded that waters at the surface have subducted and warmed, eventually meeting the definition of NASMW. It should of course be borne in mind that subpolar-origin water represents a very small proportion of the NASMW make-up.

We propose that this be clarified in the manuscript (page 13, line 14):

Our findings suggest a high-latitude source makes up a large fraction of this water (Fig. 8).
>
Our findings suggest a high-latitude source makes up a large fraction of this water, having followed a pathway from outside of our defined thermohaline range from the surface to eventually contribute to the make-up of NASMW (Fig. 8).

14. *Figure 10: This bar chart is somewhat confusing because the blue bars raise higher than the striped ones. I assume they are individually normalised? Would it not make more sense to put them on the same y-scale?*

We agree that the bar chart seems unintuitive as it is currently captioned, and so have modified the caption to explain more clearly. Normalising by the total does not show the distinct behaviour of the two reservoirs as clearly, as the volume of NASMW below the mixed layer depth is comparably small.

Probability distribution of NASMW age (hatched bars) and NASMW age below the mixed layer depth (blue bars).
>
Probability distribution of NASMW age (hatched bars) and NASMW age of NASMW restricted to be found below the mixed layer depth (blue bars).

15. *page 16, line 3: There is plenty of evidence that NADW is not formed annually, but only in some years. That the model does have annual formation clearly is a bias. This should be clearly stated*

These experiments do not display this behaviour due to the forcing applied, which is repeated, rather than historical, which allowed for general behaviours to be studied, and for long (multi-century) experiments to be conducted. We suggest the following clarification in the text:

The outcrop oscillates between the two regions with the seasonal cycle.
>
The outcrop oscillates between the two regions with the seasonal cycle. While observed NADW is known to form only in extreme winters and has a strong interannual signature (Avsic et al., 2006), our use of repeated forcing implies that formation has little year-to-year variation.

16. *page 23, line 26: So where is the rest of the NADW, if it doesn't account for 100%?*

The rest of NADW remains in the ocean. The overall budget is close, only the ventilation source are not entirely determine. For any water mass studied in this way, it is not possible to close the source budget entirely (although we are of course closer with NASMW). We have accounted for a vast majority of NADW source and chose to cap our run at 400 years due to technical constraints. It may take many more centuries to approach the proximity to closure exhibited by the source NASMW.
We propose the addition of a comment in the manuscript to stress this:

It should be noted that the mean age of model NADW is likely slightly higher, as the results detailed here do not quite account for 100% of the total budget
>
It should be noted that the mean age of model NADW is likely slightly higher. As with any water mass studied in this manner, a proportion will always remain in the ocean (closing the overall budget). Due to this, ventilation does not quite account for 100% of the total budget rererduring our 400-year run. We limit our runs to this length due to technical constraints.

17. *page 25, line 16 (and other places): here, the authors present their model results as if they realistically describe the ocean physics. However, with such a low resolution and known biases, I would prefer to see many more placements of caveats like 'within this model's context' etc*

We agree with the referee's comment and seek to enforce again our position of demonstrating a use-case of the tool, which replicates well observed behaviour of two interesting water masses. We have taken care to refer to "model NASMW" throughout, for example, and will revisit parts of the manuscript where this is not clear. We propose the following edits:

Here, we define NASMW as > We define model NASMW as (page 9, line 21)

The outcrop area of NASMW > The outcrop area of NASMW in the model (page 9, line 27)

As outlined in Sect. 2.2, we begin by identifying all NASMW > As outlined in Sect. 2.2, we begin by identifying all water matching our NASMW definition (page 10, line 2)

Newly ventilated NASMW > Newly ventilated model NASMW (page 10, line 4)

NASMW is short-lived. > Our NASMW is short-lived. (page 10, line 9)

Of that which remains in the ocean, 90% is transformed and no longer fits the definition of NASMW > Of that which remains in the ocean, 90% is transformed and no longer fits our definition of NASMW (page 10, line 10)

However, it can be observed that only around 5% of all NASMW re-ventilation occurs within its surface outcrop > However, it can be observed that only around 5% of all model NASMW re-ventilation occurs within its surface outcrop (page 11, line 1)

To track existing NASMW > To track existing model NASMW (page 11, line 9)

when NASMW volume is at its maximum > when model NASMW volume is at its maximum (page 11, line 12)

neighbourhood of the NASMW outcrop > neighbourhood of the model's NASMW outcrop (page 11, line 18)

The 60-year mean formation location of NASMW is also within this region > The 60-year mean formation location is also within this region (page 11, line 18)

corresponding to the likelihood that NASMW is found in that region > corresponding to the likelihood that model NASMW is found in that region (Figure 6, caption)

For NASMW over 5 years old, current patterns begin to have a distinct influence on formation. There is a clear signature of the Gulf Stream on the youngest NASMW > For water over 5 years old, current patterns begin to have a distinct influence on formation. There is a clear signature of the Gulf Stream on the youngest model NASMW (page 12, line 1)

The signature of the Mediterranean outflow is particularly strong on the very oldest NASMW > The signature of the Mediterranean outflow is particularly strong on the very oldest model NASMW (page 12, line 4)

when the sources of NASMW are viewed in TS space > when the model's sources of NASMW are viewed in TS space (page 12, line 6)

is not, in fact, the dominant origin of NASMW > is not, in fact, the dominant origin of NASMW in the model (page 12, line 9)

Evolution of NASMW in TS space > Evolution of model NASMW in TS space (Figure 7, caption)

We also consider the time scales involved with NASMW formation > We also consider the time scales involved with NASMW formation in the model (page 13, 3)

the expected age of the NASMW > the expected age of NASMW in the model (page 13, line 9)

NASMW as a whole > model NASMW as a whole (page 13, line 11)

within NASMW > within the model's NASMW (page 13, line 12)

Surface origins of NASMW > Surface origins of model NASMW (Figure 8, caption)

Surface origins of NASMW > Surface origins of model NASMW (Figure 9, caption)

of NASMW age > of model NASMW age (Figure 10, caption)

water of this TS signature persistently outcrops > water of this TS signature persistently outcrops in the simulation (page 15, line 1)

in the NADW TS class > in our NADW TS class (page 16, line 3)

The Greenland Sea NADW outcrop > The model's Greenland Sea NADW outcrop (page 16, 4)

subpolar-outcropping NADW > subpolar-outcropping model NADW (Figure 11,12,13,14 caption)

It is known to mix into Circumpolar Deep Water > observed NADW is known to mix into Circumpolar Deep Water (page 17, line 3)

surface-borne NADW > surface-borne model NADW (page 18, line 9)

with those of the NADW > with those of the model NADW (page 19, line 1)

Arctic-outcropping NADW > Arctic-outcropping model NADW (Figure 15,16, caption)

part of subsurface NADW. > part of subsurface NADW in the simulation. (Page 20, line 1)

The little tracer which stays in the NADW class > The little tracer which stays in our defined NADW class (Page 21, 1)

(following water from each of the two distinct NADW outcrop regions) > (following water from each of the two distinct NADW outcrop regions in the simulation) (page 21, line 3)

Hence, any surface origins of NADW in the Arctic are likely found outside of the NADW TS class > Hence, any surface origins of NADW in the model's Arctic are likely found outside of the NADW TS class (page 21, line 4)

Using this approach, 86% of the NADW budget > Using this approach, 86% of the model's NADW budget (page 21, line 8)

total volume of NADW > total volume of model NADW (page 21, line 10)

NADW spanning the entire 400-year run > Modelled NADW spanning the entire 400-year run (page 23, line 5)

NADW source waters > model NADW source waters (page 23, line 10)

The dominant TS class associated with NADW formation > The dominant TS class associated with NADW formation in the model (page 23, line 13)

of NADW formation in 400 years > of modelled NADW formation in 400 years (page 23, line 16)

Arctic NADW originates > modelled Arctic NADW originates (page 23, line 19)

and fresher than NADW > and fresher than we define NADW to be. (page 23, line 20)

of NADW formation > of simulated NADW formation (page 23, line 24)

in the Labrador and Irminger seas > in the model's Labrador and Irminger seas (page 23, line 30)

We have shown, for example, that on average > We have shown, for example, that, in our simulation, on average (page 25, line 11)

We have estimated the average age of NASMW > We have estimated the average age of model NASMW (page 25, line 14)

For NADW > For simulated NADW (page 25, line 17)

to NADW formation > to simulated NADW formation (page 25, line 19)

For NADW formed in the subpolar outcrop of the Labrador Sea > For NADW formed in the subpolar outcrop of the model's Labrador Sea (page 25, line 24)

the NADW annual outcrop > the simulated NADW annual outcrop (page 26, line 7)

18. *page 26, line 15: Long runs are not necessary for NEMOTAM to work, but only for the applications chosen here right? If the authors would have focussed on other applications, they could have used shorter runs?*

We agree that this is a poorly-written line, and have revised it for clarity:

The main barrier to higher resolution for users of our development is the  necessity of long trajectory outputs, which are required for NEMOTAM operation (Vidard et al., 2015).
>
The main barrier to higher resolution for users of our development is the necessity of frequent output and storage of trajectory snapshots from the nonlinear model, which are required for NEMOTAM operation (Vidard et al., 2015). Future versions of our tool will allow regional trajectory storage to overcome this barrier.

19. *page 26, line 18: How about 'looping' the NEMO fields? Would that work?*

This is a very interesting question and one we have considered several times. It could both work and be easily implemented (this technique has in fact been used in other studies in the NEMOTAM predecessor, c.f. Sévellec and Fedorov, 2016, J. Clim.). We chose at this resolution to run for 400 continuous years as it was cleaner and we had the necessary storage available. However, with looped fields, particularly at higher resolution, shocks (for instance from the spurious appearance and disappearance of eddies at the looping point) would introduce physical inconsistencies in the results.

---

## Author Response (AR1)

**Authors' response to reviews of the manuscript "Tracking water masses using passive-tracer transport in NEMO v3.4 with NEMOTAM: application to North Atlantic Deep Water and North Atlantic Subtropical Mode Water" by D. Stephenson, S. Müller, and F. Sévellec**

We are grateful to the editor and two referees for their positive and constructive feedback, which has been of great benefit to the work presented in this manuscript. We address their specific comments in detail below

Font legend:
*1. Reviewers' comments are in black and italic.*
2. Our responses are in black and normal font.
3. Text which has been removed from the manuscript is in red and normal font.
4. Text which has been added to the manuscript is in blue and normal font.
5. Unchanged text in the manuscript is in green and normal font.
Page references (page X, line Y) refer to the marked-up version appended to this document.

**Anonymous Referee 1**

**General comments:**

*"Stephenson et al. describe a promising and interesting addition to the toolbox for visualizing flow in ocean models. This tool is available only within the context of a tangentlinear and adjoint model but it has an advantage over the more commonly used method of Lagrangian particle tracking: insensitivity to the number of particles simulated. The authors then illustrate the types of results possible with their tool with two well known water masses: North Atlantic Subtropical Mode Water and North Atlantic Deep Water. These results are broadly consistent with results from other models and observations. Finally, as an oceanographer whose expertise is in Lagrangian methods and the water masses discussed in this paper but \*\*not\*\* with tangent-linear and adjoint models (TAM), I found the casting of the TAM approach into a purely passive tracer framework to be exceptionally illuminating; Section 2.2 in particular helped me understand the physicality behind TAMs. In my opinion, this paper contributes both a new method for tracking water masses as well as a special, simplifying case of the TAM approach; both will be useful. I recommend this paper for publication in GMD with some minor clarifications listed below."*

We thank the referee for this positive and encouraging response. We are pleased to find that our treatment of the methodology was insightful to a modeller outside of the versatile but admittedly niche field of tangent-linear and adjoint modelling.

**Specific comments**

*1. "This paper does not explicitly compare this new method to an established baseline (Lagrangian particle tracking, standard tracer methods). In particular computational demands are cited as one reason for using this method over Lagrangian particles yet there are no run time statistics in the manuscript. In my opinion, explicit quantification of the additional computational resource overhead necessary for running the TAM and this method, and insight into how this overhead scales with model resolution, is necessary in order for other scientists to fully evaluate the merits of this method relative to other methods. In the conclusion, the authors suggest it could be used with higher resolution models (i.e. ORCA12) as an "off-the-shelf" addition. That's definitely intriguing and I think more basic information about resources is necessary to support interest along these lines. Note that here I am \*not\* suggesting that the authors do any particle tracking. Rather, I think it's important to document what kinds of resources are necessary to run this method. Describing these run times will also help illustrate the actual TAM tracer workflow"*

A variation on this comment regarding runtime statistics was also brought to our attention by the second reviewer, and is an important point which we had missed. Thank you to both reviewers for

the recommendation, which we have now addressed using a number of efficiency experiments at laminar and eddying resolutions. This has been added to the manuscript as an additional section:

**2.4 Performance**

In order to test performance, the model was run for one-hundred days at 2° resolution (the ORCA2 configuration used in our case studies) and five days at 0.25° resolution (the ORCA025 configuration, Madec et al. 2012). Each configuration was run in four different modes (nonlinear, nonlinear with TAM trajectory production, tangent-linear with passive tracer, adjoint with passive tracer). Trajectory files were written once per day, and the linear advection scheme was the weighted-mean scheme described in Section \ref{sec:development_advection_schemes}. Each test was conducted with a range of parallelisation arrangements (16, 32, 64, 128, 256 and 512 CPU cores) and repeated 10 times to account for system variability (Table 1). The tests were conducted on a local HPC system, with 72 compute nodes, each offering two eight-core 2.6GHz processors (Intel Xeon E5-2650 v2) and 64GiB, connected by an InfiniBand QDR network.

The general order of time efficiency is consistent across all tests: the linear forward model is

Table 1. Results of performance tests for two model configurations (ORCA2, upper section and ORCA025, lower section). Top rows: trajectory storage requirements per output. Lower rows: runtime per model day for four model modes: Nonlinear with trajectory output (NLT), nonlinear without trajectory output (NL), tangent-linear (TL) and adjoint (AD). The standard deviation of ten runs is given as a ± uncertainty. Dashes indicate tests which failed due to insufficient memory.

| Cores (nodes) | | | 16 (1) | 32 (2) | 64 (4) | 128 (8) | 256 (16) | 512 (32) |
|---|---|---|---|---|---|---|---|---|
| ORCA2 | | | | | | | | |
| Trajectory size (MB/d) | | | 43.909 | 44.418 | 43.978 | 44.412 | 44.848 | 46.351 |
| Mode | NLT | Run time (s/d) | 2.09±0.20 | 1.54±0.17 | 1.23±0.17 | 1.39±0.26 | 2.53±0.27 | 4.08±0.35 |
| | NL | | 1.98±0.17 | 1.37±0.10 | 1.10±0.09 | 0.99±0.08 | 1.85±0.36 | 2.18±0.70 |
| | TL | | 0.78±0.01 | 0.50±0.01 | 0.36±0.09 | 0.36±0.13 | 0.45±0.18 | 0.88±0.10 |
| | AD | | 0.89±0.01 | 0.61±0.02 | 0.53±0.04 | 0.50±11 | 0.59±0.05 | 1.99±0.61 |
| ORCA025 | | | | | | | | |
| Trajectory size (MB/d) | | | | 5217.427 | 5218.495 | 5220.824 | 5217.658 | 5218.448 |
| Mode | NLT | Run time (s/d) | - | 1091.55±69.38 | 692.98±21.38 | 428.36±13.76 | 302.18±30.55 | 143.60±12.73 |
| | NL | | | 1065.48±54.56 | 656.94±13.74 | 386.34±13.89 | 262.72±34.56 | 131.70±13.28 |
| | TL | | | | 315.26±26.17 | 207.82±14.43 | 136.42±13.00 | 64.70±4.20 |
| | AD | | | - | 403.74±36.18 | 252.36±14.17 | 181.12±7.70 | 92.40±6.26 |

between 2.5 (16 cores) and 3.1 (64 cores) times as fast as the nonlinear model without trajectory output in ORCA2, and 1.9 (64 cores) to 2.0 (512 cores) times as fast in ORCA025. The adjoint is slightly less efficient. The nonlinear runs which produce trajectory output are the slowest across the board, due to the high level of output. Memory use appears higher in the linear model than the nonlinear, such that the linear model could not be run on two 64GiB compute nodes alone in ORCA025 (column two).

The model shows good scaling in the ORCA025 configuration at all CPU arrangements tested here, but begins to worsen in ORCA2 beyond 128 CPU cores. Further, the added time required for trajectory output in nonlinear ORCA2 runs on 128 CPU cores can be considerable. This is possibly due to the generation of a very large number of files during the model run, as the number of files generated is proportional to the number of CPU cores, the trajectory write frequency, and the run length. The required storage size remains relatively constant for runs with larger number of CPU cores, despite the greater number of files (Table 1, first row).

An additional limitation which we discovered during our longest experiments comes from system file-number limits, which can readily be reached for typical systems for very long runs using many cores.

> *2. Page 2: near line 25: "along with the ability to re-use a single "trajectory" run of a nonlinear model, offer an advantage over passive tracer tracking in a nonlinear OGCM."*
> *How exactly is this re-use different from re-using existing output from an eddy-resolving OGCM for different particle tracking studies? (For example, already cited in the manuscript,*

*Gary et al. (2011, 2014) and Burkholder and Lozier (2011) used different particle tracks from the same model output in three separate papers.)*

We agree with the reviewer that similar offline methods such as Lagrangian particle tracking can indeed reuse identical model outputs for distinct particle release experiments. Our comment is perhaps unclearly worded (having followed a comment regarding Lagrangian methods) but refers instead to online passive tracer methods. We have implemented the following change:

However, due to its probablistic nature, it offers an advantage over the more conventional tracking technique of Lagrangian particle modelling. Within the Lagrangian framework, ocean sensitivity to initial conditions means that a very large ensemble of initially close particle deployments may be required to representatively sample the full space of particle trajectories. A TAM can be exploited to bypass this requirement, producing a continuous probability distribution of all possible particle trajectories. As such, the method does not describe particle locations and deterministic trajectories, as a Lagrangian approach would, but tracer concentrations and probabilitic pathways. The adjoint and tangent-linear of the model can respectively track the origins and fate of passive-tracer-tagged water in this manner. This native ability to track water both forward and backward in time, along with the ability to re-use a single ``trajectory'' run of a nonlinear model, offer an advantage over passive tracer tracking in a nonlinear OGCM.
>
However, they offer two distinct advantages over online passive-tracer tracking in a nonlinear OGCM. Firstly, as with many Lagrangian methods, the TAM propagates tracer fields offline, meaning only one (more computationally demanding) simulation in the nonlinear model is required to conduct many experiments under similar conditions. Secondly, the TAM propagates probabilities both backwards and forwards in time (detailed further in Section 2.1). The adjoint and tangent-linear of the model can respectively track the origins and fate of passive-tracer-tagged water in this manner. (page 2, line 28)

*3. Page 4, near line 15: The Bower et al. 2009 and 2011 studies did not use profiling floats, only RAFOS floats, which did not profile*

Thank you to the referee for bringing this to our attention. We have corrected as follows:

However, new data from profiling floats and modelled Lagrangian drifters
>
However, new data from floats and modelled Lagrangian drifters (page 3, line 54)

*4. Page 6, near line 10: "Below the surface, it occupies a narrow latitudinal band at depths of up to 240 m (Fig. 2, red shading)." This maximum depth of 240 m is inconsistent with the greater maximum depths reported on page 9 near line 30. Perhaps the 240 m is the time mean maximum depth or a thickness? Please clarify.*

Thanks to the reviewer for highlighting this inconsistency. The value quoted does indeed refer to thickness. The correct value is in fact 310m, and has been corrected below. This also revealed an error in the y-axis tick labels of Figure 2, which has now also been corrected.

Below the surface, it occupies a narrow latitudinal band at depths of up to 240m
>
Below the surface, it occupies a narrow latitudinal band at depths of up to 310m (page 4, line 73)

*5. Page 6, near line 15, "Additionally, eddy-induced advective velocities, although present in the nonlinear trajectory, were not included for our passive tracers in NEMOTAM." Are these velocities the same as a Gent-McWilliams (GM90) eddy parameterization? If they were used in the model trajectory, why were they not used to move the tracers as well?*

As the referee supposes, these are the GM90 bolus velocities. They are by default in NEMOTAM 3.4 not used for tracer advection. In particular, the nonlinear model calculates them online, and does not explicitly store them in the trajectory, so that they are not seen by NEMOTAM. Instead, the iso-neutral slopes used for their computation are stored, (so EIV can, in fact, be retrieved). We have since, for a different adjoint study, enabled online EIV calculation using these slopes in NEMOTAM by calling the corresponding routine from the nonlinear model. This has allowed us to test the impact on passive-tracer evolution. Preliminary tests reveal little difference for the case studies described in the manuscript (Figs. A,B,C,D,E), but we have added the option of EIV to the source code for further studies.

[Figure]

*Fig A: Remaining volume of tangent-linear experiments additionally showing normalised absolute upper limit of error. Dark, dashed lines show the volume of tracer remaining without EIV. Dark solid lines are the equivalent with EIV. Faint lines show worst-case limits: EIV-on volume, +/- the globally integrated absolute volume error in every grid cell. All curves are normalised by the injected tracer volume. (Note that blue lines overlap, because of the small sensitivity of this experiment to EIV.)*

[Figure]

[Figure]

*Fig B: Snapshots of depth-integrated volume for tangent-linear NASMW propagation without (top row) and with (middle row) eddy-induced velocities at 1y (left column) 5y (centre column) and 10y (right column). The bottom row shows the difference between the two. (Note the smaller colorbar extent of the last column, demonstrating that the difference is roughly one order of magnitude smaller than the results.)*

[Figure]

*Fig C: As in Figure B, but for subpolar-outcropping NADW. (Note smaller colorbar extent of the last column, demonstrating that the difference is two orders of magnitude as small as the results.)*

[Figure]

*Fig. D: As in Fig. C, but for Arctic-outcropping NADW. (Note smaller colorbar extent of the last column, demonstrating that the difference is two orders of magnitude as small as the results.)*

[Figure]

Formation volume, 0-10 years

*Fig E: Ventilated tracer volume of NASMW (left) and NADW (right) over the first ten years of case study experiments, with (top row) and without (middle row) eddy-induced velocities. The bottom row shows the difference. (Note smaller colorbar extent of the last column, demonstrating that the difference is two orders of magnitude as small as the results.)*

We have also implemented the following change to the manuscript:

Additionally, eddy-induced advective velocities, although present in the nonlinear trajectory, were not included for our passive tracers in NEMOTAM.
>
*Additionally, eddy-induced advective velocities are computed online by the nonlinear model and not stored explicitly in the trajectory. They may be recomputed from other trajectory variables, but this is not the default behaviour of NEMOTAM. As such, they were not included for the passive-tracer experiments in this study. They have been since been enabled in the passive-tracer source code as an optional online calculation. A preliminary comparative test suggests that errors are minor for our case studies (positive and negative concentration differences tend to cancel out, but even summing absolute differences, we do not exceed ~6% relative to the injection volume). We expect the difference to be greater in steep isopycnal regions such as the Southern Ocean.* (page 5, line 8)

> *6. Page 12, near line 5, "This suggests that the NASMW surface outcrop is not, in fact, the dominant origin of NASMW...". I am confused by the use of "dominant" in this statement because it seems to me that it is not consistent with Fig. 8a which shows that a significant fraction of the NASMW (up to 70%) comes from the outcrop region. Granted, Fig. 8a is on a 0-5 year time scale and other panels in Fig. 8 clearly show sources from outside the*

*outcrop region on longer time scales, but the contributions of those other sources outside the outcrop region is still less than half of all the NASMW. Please clarify*

We thank the referee for highlighting this. We agree that Fig. 8a gives the impression that the majority of NASMW comes from the outcrop region, and have now clarified this inside the text. Unlike in the forward model, where spatial maps represent snapshots, the adjoint requires time-integrated fields be shown. As such we did not include a single outcrop contour for this figure as we thought that this would be misleading given the strong seasonal behaviour of the outcrop (it is non-existent for half of the year). It is thus difficult to convey whether or not tracer ventilated within or outside of the outcrop with this figure, for which a TS histogram is more appropriate (Fig. 9). In our TS budget, it is more clear that the dominant origin indeed has a warmer signature than surface water within the outcrop.

Our modifications are as follows:

Surface origins of NASMW as determined by the backtracking budget analysis (adjoint model simulation). Shading indicates probability density. This corresponds to the likelihood that NASMW is formed in a given region during the time periods [0,5]~yr, (5,10]~yr, (10,30]~yr and (30,50]~yr. Inset percentages show the global integral (that is, the total proportion of the budget which is formed during each time period).

Inset percentages show the global integral (that is, the total proportion of the budget which is formed during each time period).
>
Inset percentages show the global integral (that is, the total proportion of the budget which is formed during each time period). Note that, contrary to Fig. 6, which displays instantaneous fields, here time-integrated fields are displayed. Due to the large variability of its extent over the integration window, the outcrop region is not shown. (Fig. 8, caption)

–

This suggests that the NASMW surface outcrop is not, in fact, the dominant origin of NASMW, contrary to the intuition provided by laminar models of the ventilated thermocline (Luyten et al., 1983), and that ocean dynamics play a fundamental role in mode water formation.
>
While the surface distribution suggests that a relatively small neighbourhood dominates the formation of model NASMW (Fig. 8), we remind that the seasonal variability of surface properties (particularly the outcrop area) reflects strongly on the thermohaline properties of NASMW at formation (Fig. 9).  This suggests that the NASMW surface outcrop is not, in fact, the dominant origin of NASMW in the model, contrary to the intuition provided by laminar models of the ventilated thermocline (Luyten et al., 1983). The implication is that mode water formation is not a passive process, and that ocean dynamics must play a fundamental role. (page 9, line 62)

**Anonymous Referee 2**

**General comments:**

*"In this manuscript, the authors present an analysis of the pathways of NADW and NASMW in the NEMOTAM model, focussing on the long time scales and basin-scale pathways in their 2degree resolution simulation. While the manuscript is well-written and easy to follow, I am not entirely sure it fits the scope of GMD in its present form. Most of the manuscript is about the physical interpretation of the water mass pathways, and there is relatively little information about the technical details of the implementation. On the other hand, the low resolution (even for climate models) means that it is unclear how representative/relevant the results are to the physical oceanography community. Since there is hardly any comparison to observational evidence of the pathways (CFCs? C14 dating?), I suspect the*

*manuscript in its present form would raise questions in a physical oceanography journal too. I think, however that it should be feasible to rewrite the manuscript to a more traditional GMD manuscript for the audience here to find it interesting. In particular, I think the authors would then need to:"*

We thank the reviewer for their suggestions, which we address individually below. We remark that our decision to submit this manuscript to GMD was based on perceived benefit to the community of the tool (especially given the existence of the NEMO special issue), rather than to establish new ground in the analysis of either of the considered water masses. These case studies were chosen to showcase the versatility of the technique, and highlight the possibilities available to a scientist undertaking a dedicated study at higher resolution.

> *1. Add much more information about the technical details of the model. What is its memory/cpu usage? How does it scale?*

We are glad that this has been brought to our attention by both reviewers and express our gratitude again for highlighting this oversight. We have prepared an additional section to be added to the manuscript:

**2.4 Performance**

In order to test performance, the model was run for one-hundred days at 2° resolution (the ORCA2 configuration used in our case studies) and five days at 0.25° resolution (the ORCA025 configuration, Madec et al. 2012). Each configuration was run in four different modes (nonlinear, nonlinear with TAM trajectory production, tangent-linear with passive tracer, adjoint with passive tracer). Trajectory files were written once per day, and the linear advection scheme was the weighted-mean scheme described in Section 2.3. Each test was conducted with a range of parallelisation arrangements (16, 32, 64, 128, 256 and 512 CPU cores) and repeated 10 times to account for system variability (Table 1). The tests were conducted on a local HPC system, with 72 compute nodes, each offering two eight-core 2.6GHz processors (Intel Xeon E5-2650 v2) and 64GiB, connected by an InfiniBand QDR network.

The general order of time efficiency is consistent across all tests: the linear forward model is

**Table 1.** Results of performance tests for two model configurations (ORCA2, upper section and ORCA025, lower section). Top rows: trajectory storage requirements per output. Lower rows: runtime per model day for four model modes: Nonlinear with trajectory output (NLT), nonlinear without trajectory output (NL), tangent-linear (TL) and adjoint (AD). The standard deviation of ten runs is given as a ± uncertainty. Dashes indicate tests which failed due to insufficient memory.

| Cores (nodes) | | | 16 (1) | 32 (2) | 64 (4) | 128 (8) | 256 (16) | 512 (32) |
|---|---|---|---|---|---|---|---|---|
| | | | | | ORCA2 | | | |
| Trajectory size (MB/d) | | | 43.909 | 44.418 | 43.978 | 44.412 | 44.848 | 46.351 |
| Mode | NLT | Run time (s/d) | 2.09±0.20 | 1.54±0.17 | 1.23±0.17 | 1.39±0.26 | 2.53±0.27 | 4.08±0.35 |
| | NL | | 1.98±0.17 | 1.37±0.10 | 1.10±0.09 | 0.99±0.08 | 1.85±0.36 | 2.18±0.70 |
| | TL | | 0.78±0.01 | 0.50±0.01 | 0.36±0.09 | 0.36±0.13 | 0.45±0.18 | 0.88±0.10 |
| | AD | | 0.89±0.01 | 0.61±0.02 | 0.53±0.04 | 0.50±11 | 0.59±0.05 | 1.99±0.61 |
| | | | | | ORCA025 | | | |
| Trajectory size (MB/d) | | | | 5217.427 | 5218.495 | 5220.824 | 5217.658 | 5218.448 |
| Mode | NLT | Run time (s/d) | - | 1091.55±69.38 | 692.98±21.38 | 428.36±13.76 | 302.18±30.55 | 143.60±12.73 |
| | NL | | | 1065.48±54.56 | 656.94±13.74 | 386.34±13.89 | 262.72±34.56 | 131.70±13.28 |
| | TL | | | | 315.26±26.17 | 207.82±14.43 | 136.42±13.00 | 64.70±4.20 |
| | AD | | | | 403.74±36.18 | 252.36±14.17 | 181.12±7.70 | 92.40±6.26 |

between 2.5 (16 cores) and 3.1 (64 cores) times as fast as the nonlinear model without trajectory output in ORCA2, and 1.9 (64 cores) to 2.0 (512 cores) times as fast in ORCA025. The adjoint is slightly less efficient. The nonlinear runs which produce trajectory output are the slowest across the board, due to the high level of output. Memory use appears higher in the linear model than the nonlinear, such that the linear model could not be run on two 64GiB compute nodes alone in ORCA025 (column two).

The model shows good scaling in the ORCA025 configuration at all CPU arrangements tested here, but begins to worsen in ORCA2 beyond 128 CPU cores. Further, the added time required for trajectory output in nonlinear ORCA2 runs on 128 CPU cores can be considerable. This is possibly due to the generation of a very large number of files during the model run, as the number of files generated is proportional to the number of CPU cores, the trajectory write frequency, and the run length. The required storage size remains relatively constant for runs with larger number of CPU cores, despite the greater number of files (Table 1, first row).

An additional limitation which we discovered during our longest experiments comes from system file-number limits, which can readily be reached for typical systems for very long runs using many cores.

> *2. Add more information about the implementation of the model. What exactly is meant with a 'perturbation' (page 4, line 28)? A perturbation to what? How does the result depend on the choice of non-perturbed state? Is that irrelevant because of the assumption of linearity? How good is this assumption of linearity anyways? When and where does it break down? How big are the resulting errors?*

It was not our intention to provide an in-depth description of the TAM framework in a general context outside of our application. The passive tracer approach is a simplification of an otherwise quite technical method (TAM) for which there is an extensive literature. Our hope was to focus on this simplification motivated by its potential for accessibility and interdisciplinary use. We have now clarified this from the outset, and thank the reviewer for bringing this to our attention. Our modification is as follows:

These relationships are derived in full by Errico (1997). Furthermore, we have(...)
>
These relationships are derived in full by Errico (1997), who provides a thorough treatment of the linear method and its limitations, beyond the simplified, inherently linear use-case of the present study. Finally, we have(…) (page 4, line 4)

Regarding the dependency on the non-perturbed state (the trajectory), in the passive tracer context the perturbation (or as we prefer, "injection") does not actually change anything, and so can be seen as an analysis of the trajectory. In this sense, it is entirely dependent on it. The assumption of linearity and window of validity for a TAM is an example of a typical complication (c.f. Errico et al., 1993, Tellus), but one which is not an issue in the passive tracer case, as there are no feedbacks. For instance, static instability induced by a decrease in temperature, due to the high degree of nonlinearity, would not be triggered by an active linear perturbation, which is an error of the active linear model. However, in a passive linear model, no response is ever triggered by the tracer, and so these errors cannot arise. As these problems are not present in the simplified case of passive tracer tracking, we choose not to describe them, as they lie outside of the realm of our study.

> *3. Add much more validation of the model implementation. Do the TAM results indeed agree qualitatively with a full (nonlinear) tracer experiment in the 'normal' model? And how would this change when changing resolution?*

This is an interesting question in that, despite there being no effective difference between a linear and a nonlinear passive tracer, there remain some important simplifications in the TAM. In particular, the advection scheme is linearised (such that the adjoint is consistent). The nonlinear advection scheme (TVD) is considered in Section 2.3. As described (page 5, lines 4-23) some additional parameterisations are also absent in the linear advection-diffusion routines, particularly EIV. This is the native behaviour of NEMOTAM, but we have since changed it for another adjoint study, allowing us to conduct some tests on passive-tracer propagation. Preliminary tests reveal little difference for the case studies described in the manuscript, but we have added EIV as a switch in the source code for subsequent studies. (Note that we anticipate that in specific regions such as the Antarctic Circumpolar Current, which is much more intense in terms of mesoscale

eddy turbulence, this will make a larger difference.) Regarding this limitation, we have modified the manuscript to discuss it:

Additionally, eddy-induced advective velocities, although present in the nonlinear trajectory, were not included for our passive tracers in NEMOTAM.
>
Additionally, eddy-induced advective velocities are computed online by the nonlinear model and not stored explicitly in the trajectory. They may be recomputed from other trajectory variables, but this is not the default behaviour of NEMOTAM. As such, they were not included for the passive-tracer experiments in this study. They have been since been enabled in the passive-tracer source code as an optional online calculation. A preliminary comparative test suggests that errors are minor for our case studies (positive and negative concentration differences tend to cancel out, but even summing absolute differences, we do not exceed ~6% relative to the injection volume). We expect the difference to be greater in steep isopycnal regions such as the Southern Ocean. (page 5, line 8)

**Specific comments**

> *1. page 1, line 2: Be more specific what is 'probabilistic' about the tool. Is it the diffusive component?*

This question stimulated an interesting discussion for which we thank the reviewer. In the broadest sense, the Eulerian frame of reference is what determines the probabilistic nature of the approach. Parameterised diffusion, offering sub-grid-scale closure, is an important component of this, but does not determine the probabilistic nature of the approach alone. Rather, in terms of the equations themselves, the advection and diffusion of a concentration in the Eulerian framework rather than of a particle in the Lagrangian perspective, is what allows us to produce a continuous probability distribution natively. This is in opposition to reconstructing one from a series of discrete trajectories, as in the Lagrangian framework. While we choose to maintain the current wording in the abstract for brevity, we have elaborated further in the Introduction:

Lagrangian particles, conversely, are indivisible nodules which may only be advected by the immediate flow. As such, an infinite number of them is required to fully represent an equivalent dye concentration.
>
Lagrangian particles, conversely, are indivisible nodules which may only be advected by the immediate flow. The Eulerian perspective of the passive tracer method makes it inherently probabilistic - the continuous field (tracer concentration) propagated by the model corresponds to a continuous probability distribution. Lagrangian particle trajectories can be seen as discrete samples from this distribution, and so a very large (theoretically infinite) number of them is required to adequately reconstruct it. (page 2, line 14)

> *2. page 1, line 5: Be clear that the fact that tracer is removed upon contact with the surface is a choice?*

Thanks to the referee. This was previously unclear and has been rewritten.

Upon contact with the surface, the tracer is removed from the system, and a record of ventilation is produced.
>
To represent surface (re-)ventilation, we optionally decrease the tracer concentration in the surface layer, and track this concentration removal to produce a ventilation record. (Abstract)

> *3. page 2, line 1: 'bijective' is a not very common word. Explain?*

While we agree that the word is perhaps uncommon in oceanography, it has a strict mathematical definition (a one-to-one relationship) which is an important aspect of the early mathematical diagnoses of subduction described in this line. As such, we choose to maintain its use.

> *4. page 2, line 8: How many of these floats (order of magnitude) have been deployed?*

This sentence has been modified for clarity:

Despite the number of these floats, these trajectories remain under-sampled.
>
Despite their number (typically several tens of floats for dedicated pathway tracking studies), these trajectories remain under-sampled. (page 1, line 27)

> *5. page 2, line 15 (and later): it is not true that an 'infinite' number of virtual particles are needed. Depending on machine precision, the tracer concentration is also not simulated to infinite accuracy.*

We thank the referee for this important technical point, which we had overlooked. Our intention was to highlight the continuous vs. discrete nature of tracer vs. particle tracking. The manuscript has been revised accordingly:

Lagrangian particles, conversely, are indivisible nodules which may only be advected by the immediate flow. As such, an infinite number of them is required to fully represent an equivalent dye concentration.
>
Lagrangian particles, conversely, are indivisible nodules which may only be advected by the immediate flow. The Eulerian perspective of the passive tracer method makes it inherently probabilistic - the continuous field (tracer concentration) propagated by the model corresponds to a continuous probability distribution. Lagrangian particle trajectories can be seen as discrete samples from this distribution, and so a very large (theoretically infinite) number of them is required to adequately reconstruct it. (page 2, line 14)

–

Our framework is rooted in concentrations and probabilities, comparable to the statistical properties of a high-resolution Lagrangian approach with infinitely many particles.
>
Our framework is rooted in concentrations and probabilities, comparable to the statistical properties of a high-resolution Lagrangian approach with a large (theoretically infinite) number of particles. (page 16, line 29)

> *6. page 2, line 24: what exactly is meant with a 'probabilitic pathway'?*

This line was modified in response to the first reviewer's second comment:

However, due to its probablistic nature, it offers an advantage over the more conventional tracking technique of Lagrangian particle modelling. Within the Lagrangian framework, ocean sensitivity to initial conditions means that a very large ensemble of initially close particle deployments may be required to representatively sample the full space of particle trajectories. A TAM can be exploited to bypass this requirement, producing a continuous probability distribution of all possible particle trajectories. As such, the method does not describe particle locations and deterministic trajectories, as a Lagrangian approach would, but tracer concentrations and probabilitic pathways. The adjoint and tangent-linear of the model can respectively track the origins and fate of passive-tracer-tagged water in this manner. This native ability to track water both forward and backward in time, along with the ability to re-use a single ``trajectory'' run of a nonlinear model, offer an advantage over passive tracer tracking in a nonlinear OGCM.

>
However, they offer two distinct advantages over online passive-tracer tracking in a nonlinear OGCM. Firstly, as with many Lagrangian methods, the TAM propagates tracer fields offline, meaning only one (more computationally demanding) simulation in the nonlinear model is required to conduct many experiments under similar conditions. Secondly, the TAM propagates probabilities both backwards and forwards in time (detailed further in Section 2.1). The adjoint and tangent-linear of the model can respectively track the origins and fate of passive-tracer-tagged water in this manner. (page 2, line 28)

> 7. page 3, line 9: The recent Bower et al review paper on Atlantic pathways (https://agupubs.onlinelibrary.wiley.com/doi/full/10.1029/2019JC015014) is not mentioned here?

While this paper is referenced elsewhere (Page 2, line 8), it describes Lagrangian pathways rather than attempts to linearly model passive tracers, which is the focus of this line.

> 8. page 5, line 2: perhaps very briefly explain bra-ket notation to oceanographers?

We agree that the notation is uncommon, and have added the following short explanation:

(...)we have used the "bra-ket" notation of Dirac (1939).

(...)we have used the "bra-ket" notation of Dirac (1939) in which row and column vectors are respectively written as bras ($<a|$) and kets ($|b>$) such that closed bra-ket terms become scalar (through the scalar product $<a|b>=c$). (page 3, line 96)

> 9. page 6, line 4: what is the effect of the non-resolvement of the ice layers in NEMOTAM? Why are these not implemented? Is it technically impossible?

This is a well-raised point which requires further elaboration. In the context of an active tracer, ice formation and melt is highly nonlinear, and represents an error in NEMOTAM's linearization approach. However, a passive tracer would not interact with these processes. Furthermore, any change in volume is handled in the trajectory, so concentration changes due to, for example, dilution, are pre-emptively addressed by the trajectory.

We have modified the manuscript as follows:

There are two ice layers Fichefet (1997) in the background state, although these are not incorporated into NEMOTAM.
>
There are two ice layers Fichefet (1997) in the background state. The ice model is not linearised or coupled to NEMOTAM, although in the context of this study, ice dynamics are inherently unnafected (as a tracer which is passive cannot affect ice) (page 4, line 59)

> 10. page 9, line 23: why not impose an Ertel PV criterion, as is very often done? Is that possible within the TAM framework?

We had considered several criteria when preparing the experiments, and PV would have indeed been possible (any property of the background state can be used to determine where to inject the tracer). Given the lack of a universally accepted NASMW definition and in the interests of simplicity, we used a thickness criterion, as in Gary et al. (2014, J. Phys. Ocean.).

Here, we define NASMW(...)
>

While the method allows us to define water masses in terms of any model variable, we choose for simplicity to utilise the common approach of a temperature-salinity range. In particular, we define model NASMW(...) (page 7, line 23)

> 11. page 10, line 12: I don't understand the meaning of the word 'mechanically' here

This appears to be a typo and has been corrected. We thank the referee for bringing this to our attention.

(...)remaining tracer mechanically tends to reach deeper, colder, fresher waters.
>
(...)remaining tracer tends to reach deeper, colder, fresher waters. (page 8, line 39)

> 12. page 13, line 5: How sure are the authors that this indeed is a lognormal distribution? I would have liked to see a goodness-of-fit analysis. There are other distributions that produce similar-looking PDFs

We agree with the referee that this description is too specific and have removed it.

This distribution resembles a log-normal PDF with a lower skewness for NASMW lying below the MLD
>
There is a visibly lower skewness for NASMW lying below the MLD (page 9, line 71)

> 13. Figure 8: I am somewhat surprised that some of the NASMW originates near Greenland? Wasn't one of the conditions that the temperature was higher than 17C? Does that occur within the model in the region near Greenland??

After injection, the tracer is not tied to any particular temperature range, or indeed water mass. Under the assumption that the pathway of the tracer-tagged water is representative of the history of NASMW, it must be concluded that waters at the surface have subducted and warmed, eventually meeting the definition of NASMW. It should of course be borne in mind that subpolar-origin water represents a very small proportion of the NASMW make-up

We have clarified this in the manuscript:

Our findings suggest a high-latitude source makes up a large fraction of this water
>
Our findings suggest a high-latitude source makes up a large fraction of this water, having followed a pathway from outside of our defined thermohaline range from the surface to eventually contribute to the make-up of NASMW (page 9, line 88)

> 14. Figure 10: This bar chart is somewhat confusing because the blue bars raise higher than the striped ones. I assume they are individually normalised? Would it not make more sense to put them on the same y-scale?

We agree that the bar chart seems unintuitive as it is currently captioned, and so have modified the caption to explain more clearly. Normalising by the total does not show the distinct behaviour of the two reservoirs as clearly, as the volume of NASMW below the mixed layer depth is comparably small.

Probability distribution of NASMW age (hatched bars) and NASMW age below the mixed layer depth (blue bars).
>
Probability distribution of NASMW age (hatched bars) and NASMW age of NASMW restricted to be found below the mixed layer depth (blue bars). (Fig. 10, caption)

*15. page 16, line 3: There is plenty of evidence that NADW is not formed annually, but only in some years. That the model does have annual formation clearly is a bias. This should be clearly stated*

These experiments do not display this behaviour due to the forcing applied, which is repeated, rather than historical, which allowed for general behaviours to be studied, and for long (multi-century) experiments to be conducted. We have made this more explicit in the text:

The outcrop oscillates between the two regions with the seasonal cycle.
>
The outcrop oscillates between the two regions with the seasonal cycle. While observed NADW is known to form only in extreme winters and has a strong interannual signature (Avsic et al., 2006), our use of repeated forcing implies that formation has little year-to-year variation. (page 10, line 9)

*16. page 23, line 26: So where is the rest of the NADW, if it doesn't account for 100%?*

The rest of NADW remains in the ocean. The overall budget is close, only the ventilation source are not entirely determine. For any water mass studied in this way, it is not possible to close the source budget entirely (although we are of course closer with NASMW). We have accounted for a vast majority of NADW source and chose to cap our run at 400 years due to technical constraints. It may take many more centuries to approach the proximity to closure exhibited by the source NASMW. We propose the addition of a comment in the manuscript to stress this:

It should be noted that the mean age of model NADW is likely slightly higher, as the results detailed here do not quite account for 100% of the total budget
>
It should be noted that the mean age of model NADW is likely slightly higher. As with any water mass studied in this manner, a proportion will always remain in the ocean (closing the overall budget). Due to this, ventilation does not quite account for 100% of the total budget during our 400-year run. We limit our runs to this length due to technical constraints. (page 16, line 8)

*17. page 25, line 16 (and other places): here, the authors present their model results as if they realistically describe the ocean physics. However, with such a low resolution and known biases, I would prefer to see many more placements of caveats like 'within this model's context' etc*

We agree with the referee's comment and seek to enforce again our position of demonstrating a use-case of the tool, which replicates well observed behaviour of two interesting water masses. We have taken care to refer to "model NASMW" throughout, for example, and will revisit parts of the manuscript where this is not clear. We have edited the manuscript as follows:

Here, we define NASMW as > We define model NASMW as (page 7, line 27)

The outcrop area of NASMW > The outcrop area of NASMW in the model (page 8, line 3)

As outlined in Sect. 2.2, we begin by identifying all NASMW > As outlined in Sect. 2.2, we begin by identifying all water matching our NASMW definition (page 8, line 18)

Newly ventilated NASMW > Newly ventilated model NASMW (page 8, line 23)

NASMW is short-lived. > Our NASMW is short-lived. (page 8, line 33)

Of that which remains in the ocean, 90% is transformed and no longer fits the definition of NASMW > Of that which remains in the ocean, 90% is transformed and no longer fits our definition of NASMW (page 8, line 36)

However, it can be observed that only around 5% of all NASMW re-ventilation occurs within its surface outcrop > However, it can be observed that only around 5% of all model NASMW re-ventilation occurs within its surface outcrop (page 8, line 42)

To track existing NASMW > To track existing model NASMW (page 9, line 13)

when NASMW volume is at its maximum > when model NASMW volume is at its maximum (page 9, line 19)

neighbourhood of the NASMW outcrop > neighbourhood of the model's initial NASMW outcrop (page 9, line 32)

The 60-year mean formation location of NASMW is also within this region > The 60-year mean formation location is also within this region (page 9, line 33)

corresponding to the likelihood that NASMW is found in that region > corresponding to the likelihood that model NASMW is found in that region (Figure 6, caption)

For NASMW over 5 years old, current patterns begin to have a distinct influence on formation. There is a clear signature of the Gulf Stream on the youngest NASMW > For water over 5 years old, current patterns begin to have a distinct influence on formation. There is a clear signature of the Gulf Stream on the youngest model NASMW (page 9, line 38)

The signature of the Mediterranean outflow is particularly strong on the very oldest NASMW > The signature of the Mediterranean outflow is particularly strong on the very oldest model NASMW (page 9, line 45)

when the sources of NASMW are viewed in TS space > when the model's sources of NASMW are viewed in TS space (page 9, line 49)

is not, in fact, the dominant origin of NASMW > is not, in fact, the dominant origin of NASMW in the model (page 9, line 58)

Evolution of NASMW in TS space > Evolution of model NASMW in TS space (Figure 7, caption)

We also consider the time scales involved with NASMW formation > We also consider the time scales involved with NASMW formation in the model (page 9, 66)

the expected age of the NASMW > the expected age of NASMW in the model (page 9, line 79)

NASMW as a whole > model NASMW as a whole (page 9, line 83)

Surface origins of NASMW > Surface origins of model NASMW (Figure 8, caption)

Surface origins of NASMW > Surface origins of model NASMW (Figure 9, caption)

of NASMW age > of model NASMW age (Figure 10, caption)

water of this TS signature persistently outcrops > water of this TS signature persistently outcrops in the simulation (page 10, line 4)

in the NADW TS class > in our NADW TS class (page 10, line 16)

The Greenland Sea NADW outcrop > The model's Greenland Sea NADW outcrop (page 10, 18)

subpolar-outcropping NADW > subpolar-outcropping model NADW (Figure 11,12,13,14 caption)

It is known to mix into Circumpolar Deep Water > observed NADW is known to mix into Circumpolar Deep Water (page 11, line 36)

surface-borne NADW > surface-borne model NADW (page 13, line 24)

with those of the NADW > with those of the model NADW (page 13, line 27)

Arctic-outcropping NADW > Arctic-outcropping model NADW (Figure 15,16, caption)

part of subsurface NADW. > part of subsurface NADW in the simulation. (page 13, line 32)

The little tracer which stays in the NADW class > The little tracer which stays in our defined NADW class (page 13, line 35)

(following water from each of the two distinct NADW outcrop regions) > (following water from each of the two distinct NADW outcrop regions in the simulation) (page 13, line 39)

Hence, any surface origins of NADW in the Arctic are likely found outside of the NADW TS class > Hence, any surface origins of NADW in the model's Arctic are likely found outside of the NADW TS class (page 13, line 43)

Using this approach, 86% of the NADW budget > Using this approach, 86% of the model's NADW budget (page 13, line 47)

total volume of NADW > total volume of model NADW (page 13, line 52)

NADW spanning the entire 400-year run > Modelled NADW spanning the entire 400-year run (page 14, line 8)

NADW source waters > model NADW source waters (page 15, line 8)

The dominant TS class associated with NADW formation > The dominant TS class associated with NADW formation in the model (page 15, line 15)

of NADW formation in 400 years > of modelled NADW formation in 400 years (page 15, line 22)

Arctic NADW originates > modelled Arctic NADW originates (page 15, line 27)

and fresher than NADW > and fresher than we define NADW to be. (page 15, line 30)

of NADW formation > of simulated NADW formation (page 16, line 4)

in the Labrador and Irminger seas > in the model's Labrador and Irminger seas (page 16, line 21)

We have shown, for example, that on average > We have shown, for example, that, in our simulation, on average (page 16, line 59)

We have estimated the average age of NASMW > We have estimated the average age of model NASMW (page 16, line 65)

For NADW > For simulated NADW (page 16, line 70)

to NADW formation > to simulated NADW formation (page 16, line 75)

For NADW formed in the subpolar outcrop of the Labrador Sea > For NADW formed in the subpolar outcrop of the model's Labrador Sea (page 16, line 86)

the NADW annual outcrop > the simulated NADW annual outcrop (page 17, line 12)

> *18. page 26, line 15: Long runs are not necessary for NEMOTAM to work, but only for the applications chosen here right? If the authors would have focussed on other applications, they could have used shorter runs?*

We agree that this is a poorly-written line, and have revised it for clarity:

The main barrier to higher resolution for users of our development is the necessity of long trajectory outputs, which are required for NEMOTAM operation (Vidard et al., 2015).
>
The main barrier to higher resolution for users of our development is the necessity of frequent output and storage of trajectory snapshots from the nonlinear model, which are required for NEMOTAM operation (Vidard et al., 2015). Future versions of our tool will allow regional trajectory storage to overcome this barrier. (page 17, line 26)

> *19. page 26, line 18: How about 'looping' the NEMO fields? Would that work?*

This is a very interesting question and one we have considered several times. It could both work and be easily implemented (this technique has in fact been used in other studies in the NEMOTAM predecessor, c.f. Sévellec and Fedorov, 2016, J. Clim.). We chose at this resolution to run for 400 continuous years as it was cleaner and we had the necessary storage available. However, with looped fields, particularly at higher resolution, shocks (for instance from the spurious appearance and disappearance of eddies at the looping point) would introduce physical inconsistencies in the results.

[revised manuscript text omitted]